# 4D printing of MXene hydrogels for high-efficiency pseudocapacitive energy storage

Ke Li [1,2] ✉, Juan Zhao[1,2], Ainur Zhussupbekova [2,3], Christopher E. Shuck [4], Lucia Hughes [1,2], Yueyao Dong [2], Sebastian Barwich[3], Sebastien Vaesen [1,2], Igor V. Shvets[3], Matthias Möbius [3], Wolfgang Schmitt[1,2], Yury Gogotsi [4] ✉ & Valeria Nicolosi [1,2] ✉

2D material hydrogels have recently sparked tremendous interest owing to their potential in diverse applications. However, research on the emerging 2D MXene hydrogels is still in its infancy. Herein, we show a universal 4D printing technology for manufacturing MXene hydrogels with customizable geometries, which suits a family of MXenes such as $Nb_2CT_x$, $Ti_3C_2T_x$, and $Mo_2Ti_2C_3T_x$. The obtained MXene hydrogels offer 3D porous architectures, large specific surface areas, high electrical conductivities, and satisfying mechanical properties. Consequently, ultrahigh capacitance (3.32 F cm$^{-2}$ (10 mV s$^{-1}$) and 233 F g$^{-1}$ (10 V s$^{-1}$)) and mass loading/thickness-independent rate capabilities are achieved. The further 4D-printed $Ti_3C_2T_x$ hydrogel micro-supercapacitors showcase great low-temperature tolerance (down to −20 °C) and deliver high energy and power densities up to 93 μWh cm$^{-2}$ and 7 mW cm$^{-2}$, respectively, surpassing most state-of-the-art devices. This work brings new insights into MXene hydrogel manufacturing and expands the range of their potential applications.

The recent boom in portable electronics, hybrid/electric vehicles, and intermittent energy (e.g., sun and wind) harvesting highlights the need for efficient energy-storage systems[1,2]. Supercapacitors (SCs), as promising candidates, have stuck out because of their high power density and long cycle life, which enable fast charging and eliminate the need for replacement of energy-storage devices over the lifetime of equipment that they power. However, SCs suffer from low energy density which impedes their wide implementations; one feasible approach to bypass this is developing advanced electrode materials[3].

Conductive hydrogels, particularly those based on conductive 2D materials (e.g., graphene and MXene), can be used as electrode materials with high energy and power densities[4,5]. They not only possess large surface areas and hydrophilic properties but also maintain the high electrical conductivity of 2D materials, allowing for electrochemical reactions, fast electrolyte ion transport and electron transfer even in thick electrodes[4]. 2D MXenes with a formula of $M_{n+1}X_nT_x$ (M is an early transition metal, X is carbon and/or nitrogen, n is an integer between 1 and 4, and $T_x$ represents surface functional groups) offer a large number of promising candidates for designing conductive 2D hydrogels owing to their large surface-area-to-volume ratios, high electrical conductivities (≥20,000 S cm$^{-1}$), redox capable surface groups, and chemical/structural diversities[6–11]. To date, several MXene hydrogels have been developed by filtration[12], or using metal ions[13], graphene oxide[14], or polymers (e.g., polyvinyl alcohol (PVA)[15], polyacrylamide[16], cellulose[17], chitosan[18], poly(acrylic acid)[19], and poly(N-isopropylacrylamide)[20]) as crosslinkers, and have

[1]Centre for Research on Adaptive Nanostructures and Nanodevices (CRANN) & Advanced Materials and BioEngineering Research Centre (AMBER), Trinity College Dublin, Dublin Dublin 2, Ireland. [2]School of Chemistry, Trinity College Dublin, Dublin Dublin 2, Ireland. [3]School of Physics, Trinity College Dublin, Dublin Dublin 2, Ireland. [4]A. J. Drexel Nanomaterials Institute and Department of Materials Science and Engineering, Drexel University, Philadelphia, PA 19104, USA. ✉e-mail: like@tcd.ie; gogotsi@drexel.edu; nicolov@tcd.ie

demonstrated some success. Nevertheless, research on MXene hydrogels is still in its infancy, and serious challenges remain. First, the previous reports all focused on $Ti_3C_2T_x$; no other MXene hydrogels (beyond $Ti_3C_2T_x$) have been reported. Second, the polymer cross-linkers employed were insulating, which lowered the electrical conductivity of MXene hydrogels and weakened their electrical/electrochemical performance. Third, the geometry of these MXene hydrogels strongly depends on the shape and size of molds, which is unlikely to meet the requirements of complexity and precision in many scenarios, especially in the context of the rapid development of portable electronics.

Additive manufacturing, or 3D printing, offers an efficient approach to realizing the precise, mold-free, and low-cost fabrication of complex objects by layer-by-layer deposition of material[21]. With the introduction of the fourth dimension of time, 4D printing (3D printing + time) emerged[22]. It not only inherits all merits of 3D printing but also allows the static objects created by 3D printing to intentionally change their shape, property, or functionality over time when exposed to specific external stimuli (e.g., heat, light, water, pH)[23], endowing the printed objects with new features. However, no related works on MXene hydrogels were ever reported.

Herein, an advanced 4D printing technology is developed for scalable manufacturing of MXene hydrogels, with the conducting polymer poly(3,4-ethylenedioxythiophene):poly(styrene sulfonate) (PEDOT:PSS) as the crosslinker. Differing from the dissolvable MXene sol patterns produced by traditional 3D printing (Supplementary Table 1), in our 4D printing technology, crosslinked MXene hydrogels with enhanced mechanical strengths are obtained by employing a simple heat-stimulated self-assembly process. This strategy shows remarkable universality, which allows the fabrication of a series of MXene hydrogels including $Nb_2CT_x$, $Ti_3C_2T_x$, and $Mo_2Ti_2C_3T_x$ hydrogels. Notably, these three kinds of MXenes possess different numbers of atomic layers ($n = 1$, 2, and 3) and different transition metals (Nb, Ti, Mo) on their surface, representing a large family of MXenes with diverse structures and properties. In addition, the geometries of the 4D-printed MXene hydrogels are precisely customizable: various complex architectures such as microlattice, rectangular hollow prism, Chinese knot, "CRANN" logo, and micro-supercapacitor (MSC) units are easily produced. Meanwhile, they all feature excellent hydrophilic properties, large specific surface areas, and high electrical

conductivities. Consequently, the $Ti_3C_2T_x$ hydrogel electrodes show a large areal capacitance of 3.32 F cm$^{-2}$ at 10 mV s$^{-1}$, an ultrahigh specific capacitance of 232.9 F g$^{-1}$ at 10 V s$^{-1}$, and mass loading/thickness-independent rate capabilities. Furthermore, the 4D-printed MXene hydrogel MSCs enable low-temperature operation, with high capacitance retentions of 90.6% at 0 °C and 82.2% at −20 °C. More importantly, the energy and power densities of our MSCs reach up to 92.88 µWh cm$^{-2}$ and 6.96 mW cm$^{-2}$, respectively, demonstrating their potential as efficient energy-storage devices.

## Results

### 4D printing of MXene hydrogels

Figure 1 schematically represents the 4D printing methodology developed for manufacturing MXene hydrogels. A homogenous ink with the required rheological properties was formulated by mixing few-layer MXenes, PEDOT:PSS, and additives (dimethyl sulfoxide (DMSO), sulfuric acid ($H_2SO_4$), and sodium L-ascorbate). After 3D printing, various MXene sol patterns and architectures were obtained, which further transformed into MXene hydrogels via a heat-stimulated self-assembly process. Additives play a crucial role during the self-assembly process, as DMSO and $H_2SO_4$ simultaneously facilitate the self-assembly of MXene hydrogels, whilst sodium L-ascorbate is a reducing agent that protects MXenes from oxidation[24].

$Nb_2CT_x$, $Ti_3C_2T_x$, and $Mo_2Ti_2C_3T_x$ MXenes were synthesized by etching their precursor MAX phases in hydrochloric acid-lithium fluoride (HCl·LiF) or hydrofluoric acid (HF) solutions (see Methods). The obtained MXenes all present characteristic peaks in the X-ray diffraction patterns (Supplementary Fig. 1) and contain ultrathin 2D flakes in the transmission electron microscopy (TEM) images (Supplementary Fig. 2), demonstrating their high quality. Since the self-assembly process is crucial for the formation of MXene hydrogels from MXene sols, this was first investigated on $Ti_3C_2T_x$ MXene as an example, before moving to 3D printing. As shown in Fig. 2a, pure PEDOT:PSS hydrogel shows the most distinct volume shrinkage after self-assembly at 90 °C for 6 h. With the continuous introduction of $Ti_3C_2T_x$ MXene, the volume contraction of hydrogels becomes progressively less prominent. Hydrogels could not be formed from pure MXene. The maximum load of $Ti_3C_2T_x$ in hydrogels is 80 wt.%, and in this case, the produced $Ti_3C_2T_x$ hydrogel could be readily transferred into another glass vial without breakage (Supplementary Fig. 3a). In addition, the

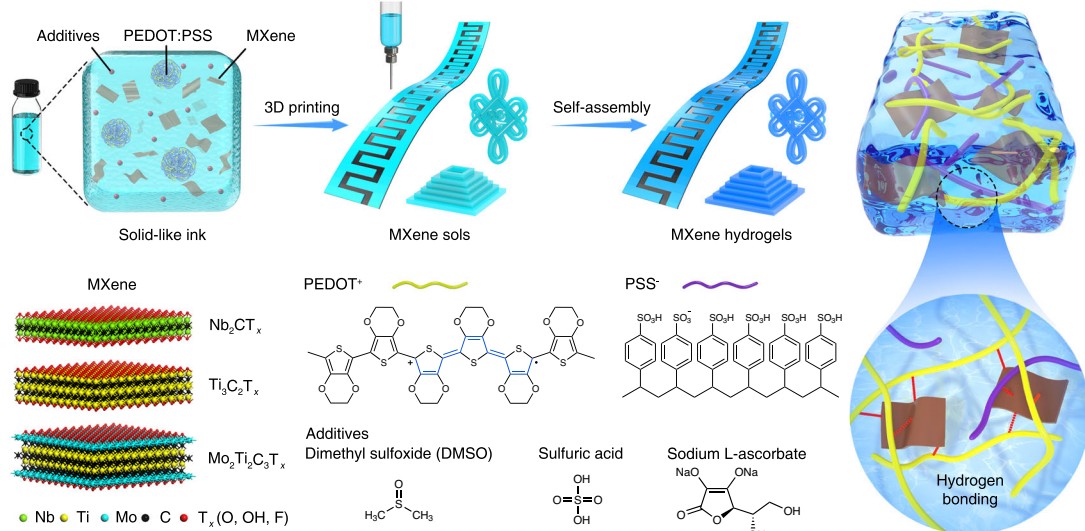

**Fig. 1 | Schematic illustration of 4D printing of MXene hydrogels.** The composite inks consisting of MXenes, PEDOT:PSS, and additives (DMSO, $H_2SO_4$, and sodium L-ascorbate) are 3D-printed into designed patterns first, followed by a self-assembly process, MXene sols transform into MXene hydrogels. Three kinds of MXenes, $Nb_2CT_x$, $Ti_3C_2T_x$, and $Mo_2Ti_2C_3T_x$, are employed for demonstration of the universality and feasibility of this technology.

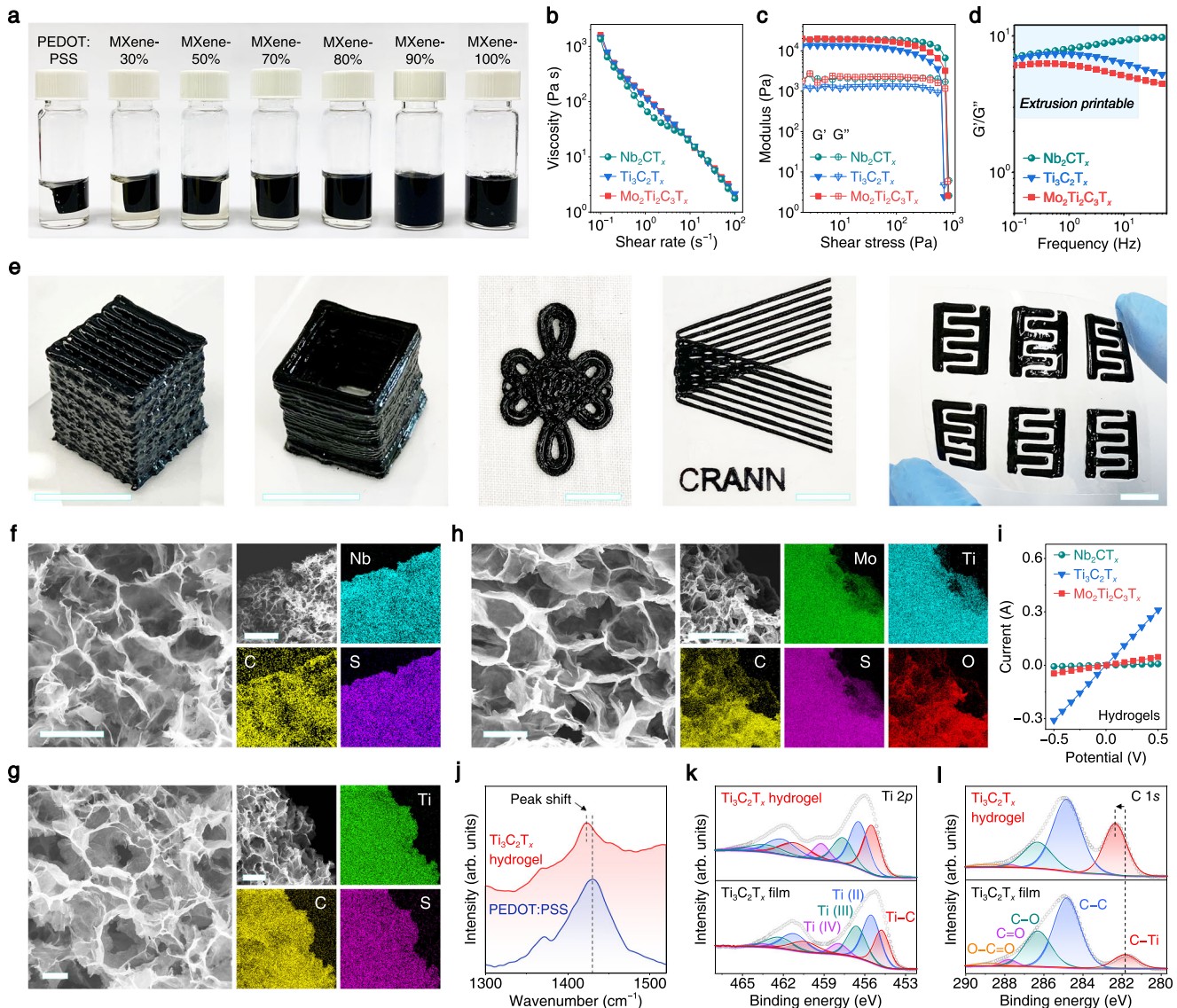

**Fig. 2 | Characterizations of MXene inks and hydrogels. a** Photographs of $Ti_3C_2T_x$ hydrogels with different MXene contents prepared by self-assembly. **b** Viscosity as a function of shear rate for $Nb_2CT_x$, $Ti_3C_2T_x$ and $Mo_2Ti_2C_3T_x$ inks. **c** Storage modulus (G') and loss modulus (G'') as a function of the shear stress for $Nb_2CT_x$, $Ti_3C_2T_x$ and $Mo_2Ti_2C_3T_x$ inks. **d** Frequency dependency of the ratio of the G' to G'' for $Nb_2CT_x$, $Ti_3C_2T_x$ and $Mo_2Ti_2C_3T_x$ inks. **e** Photographs of 4D-printed MXene hydrogel architectures (from left to right): $Ti_3C_2T_x$ hydrogel microlattice on glass slide, $Ti_3C_2T_x$ hydrogel rectangular hollow prism on glass slide, $Nb_2CT_x$ hydrogel Chinese knot on cloth, $Nb_2CT_x$ hydrogel "CRANN" logo on PET film, flexible

$Mo_2Ti_2C_3T_x$ hydrogel MSC units on PET film. All scale bars in **e** correspond to 1 cm. **f** SEM and energy-dispersive X-ray spectroscopy (EDX) mapping images of $Nb_2CT_x$ hydrogel. **g** SEM and EDX mapping images of $Ti_3C_2T_x$ hydrogel. **h** SEM and EDX mapping images of $Mo_2Ti_2C_3T_x$ hydrogel. All scale bars in SEM images in **f**–**h** are 5 μm and all scale bars in EDX mapping images in **f**–**h** are 20 μm. **i** I–V curves of $Nb_2CT_x$, $Ti_3C_2T_x$, and $Mo_2Ti_2C_3T_x$ hydrogels with a size of $10 \times 2 \times 2$ mm. **j** Raman spectra of pure PEDOT:PSS film and 4D-printed $Ti_3C_2T_x$ hydrogel. High resolution **k** Ti 2*p* and **l** C 1*s* XPS spectra of filtered $Ti_3C_2T_x$ film and 4D-printed $Ti_3C_2T_x$ hydrogel. Binding energies were all calibrated to the C 1*s* peak at 284.8 eV.

shapes of $Ti_3C_2T_x$ hydrogels (80 wt.%) are easily customizable: cones, hemispheres, cylinders, and fibers were all produced using different molds. More interestingly, the fiber-shaped $Ti_3C_2T_x$ hydrogels show great flexibility, which has been demonstrated by patterning the letters "TCD" (Supplementary Fig. 3b). Thus, in the following experiments, the mass content of MXene in hydrogels was set to 80 wt.%. Apart from the MXene mass ratio, $H_2SO_4$ and DMSO also have crucial effects on the self-assembly process (Supplementary Figs. 4 and 5). In the absence of DMSO, the $Ti_3C_2T_x$ hydrogels obtained in 0.1 M $H_2SO_4$ are rigid and fragile, and in 1 M $H_2SO_4$, hydrogels cannot even form (Supplementary Fig. 4a). While in absence of $H_2SO_4$, $Ti_3C_2T_x$ hydrogels are soft and easily broken (Supplementary Fig. 4b), the volume fraction of DMSO also matters (Supplementary Fig. 5). Ultimately, the optimal formula for additives was determined to be 0.1 M $H_2SO_4$ plus 26 vol.% DMSO,

and the molar ratio of the reducing agent sodium L-ascorbate to the metal atom in MXene was 1:1 (see Methods).

To enable 3D printing, the dispersions consisting of MXenes ($Nb_2CT_x$, $Ti_3C_2T_x$, or $Mo_2Ti_2C_3T_x$), PDEOT:PSS, and additives in the above discussed proportions, were condensed by high-speed centrifugation, resulting in solid-like inks with MXene-PEDOT:PSS presenting a concentration of ~50 mg mL$^{-1}$ (Supplementary Fig. 6). A rheological study was carried out to probe the suitability of these inks for 3D printing (Fig. 2b–d). As shown in Fig. 2b, the $Nb_2CT_x$, $Ti_3C_2T_x$, and $Mo_2Ti_2C_3T_x$ inks all exhibit high apparent viscosities and shear-thinning non-Newtonian behavior, which ensures the continuous flow of inks through nozzles and the shape fidelity once the applied stress is released. Figure 2c depicts the storage modulus (G') and loss modulus (G'') of the inks. The yield stress (crossover point between G' and G'')

values of $Nb_2CT_x$, $Ti_3C_2T_x$, and $Mo_2Ti_2C_3T_x$ inks were measured to be 803, 664, and 778 Pa, respectively, which markedly exceed the requirement for 3D printing of MXenes (ca. 100 Pa)[25,26]. Below the yield stress (G′ > G″), the three inks behave as solids, and they flow under higher shear stress (G″ > G′), which allows for ink extrusion through nozzles. The frequency scan was also performed, as displayed in Supplementary Fig. 7, the three inks all possess higher G′ than G″ with G′ and G″ being frequency-independent throughout the measured frequency range. Moreover, the ink-processability-related parameter G′/G″ ratios of three inks fall within the range for MXene extrusion printing (from ca. 2.5 to 20) (Fig. 2d)[27]. This once again demonstrates the suitability of these inks for extrusion printing. Various MXene sol architectures were produced and are shown in Supplementary Movies 1–5. After a self-assembly process, the 3D-printed MXene sols further transformed into MXene hydrogels with great shape fidelity; a $Ti_3C_2T_x$ hydrogel microlattice and a $Ti_3C_2T_x$ hydrogel rectangular hollow prism on glass slides, a $Nb_2CT_x$ hydrogel Chinese knot on cloth, a $Nb_2CT_x$ hydrogel "CRANN" logo and flexible $Mo_2Ti_2C_3T_x$ hydrogel MSC units on polyethylene terephthalate (PET) films were fabricated with this technique to demonstrate its versatility and scalability (Fig. 2e, Supplementary Fig. 8). Additionally, treatment with concentrated $H_2SO_4$ boosted the mechanical properties of the 4D-printed hydrogels. Even after vigorous shaking, the treated $Ti_3C_2T_x$ hydrogel microlattice and rectangular hollow prism retained their integrity without displaying breakage or deformation (Supplementary Movie 6, Supplementary Fig. 9). The enhanced mechanical strength will benefit MXene hydrogels in many applications. In contrast, the 3D-printed $Ti_3C_2T_x$ sol microlattice broke into fragments after shaking (Supplementary Movie 7, Supplementary Fig. 10).

The microstructures of MXene hydrogels were examined with scanning electron microscopy (SEM). As displayed in Fig. 2f–h, the $Nb_2CT_x$, $Ti_3C_2T_x$, and $Mo_2Ti_2C_3T_x$ hydrogels have porous structures with pore sizes ranging from less than a micrometer to several micrometers. TEM images also show the 3D framework of MXene hydrogel, and polymer chains sandwiched between MXene layers (Supplementary Fig. 11). $N_2$ adsorption/desorption isotherms were recorded at 77 K to further investigate their Brunauer-Emmett-Teller (BET) specific surface areas (SSAs) and pore size distributions (Supplementary Fig. 12). The results confirmed the porosity of the MXene hydrogels materials, as indicated by their type II adsorption isotherms, typical of macroporous sheet-like adsorbents, but combined with hysteresis, typical of mesoporous materials (Type IV isotherms). While $Mo_2Ti_2C_3T_x$ and $Nb_2CT_x$ hydrogels present the expected narrow hysteresis loops, the $Ti_3C_2T_x$ hydrogel presents a broader hysteresis which indicates the presence of slit-shaped mesopores[28]. The SSAs of $Nb_2CT_x$, $Ti_3C_2T_x$, and $Mo_2Ti_2C_3T_x$ hydrogels were calculated to be 48.7, 21.8, and $21\,m^2\,g^{-1}$, respectively, which are comparable to other porous MXene materials[29]. Notably, despite the porous structure inducing the formation of finite electron transport pathways, the electrical conductivities of our MXene hydrogels still reach 37, 1548, and $231\,S\,m^{-1}$ for $Nb_2CT_x$, $Ti_3C_2T_x$, and $Mo_2Ti_2C_3T_x$ hydrogels, respectively (Fig. 2i). This showcases the potential of these materials for applications in sensors, bioelectronics, electromagnetic interference shielding, electrochemical energy storage, etc. (Supplementary Table 2). The conductivity difference between these hydrogels should be ascribed to the intrinsic conductivity difference of the three MXenes (Supplementary Fig. 13).

## Probing self-assembly mechanism

To elucidate the self-assembly mechanism and the interactions between MXenes and PEDOT:PSS, Raman spectroscopy and X-ray photoelectron spectroscopy (XPS) were employed. In the Raman spectrum of the PEDOT:PSS film (Fig. 2j), the band at $1430\,cm^{-1}$ arises from the $C_\alpha = C_\beta$ stretching vibration of thiophene rings[30], while it redshifts to $1422\,cm^{-1}$ in the $Ti_3C_2T_x$ hydrogel, suggesting the conformation change from

benzene structure to quinoid structure and thus elongated conjugation lengths of $PEDOT^+$ chains[31]. In addition, the narrower $1422\,cm^{-1}$ band of the $Ti_3C_2T_x$ hydrogel compared with the PEDOT:PSS film also confirms the increased crystallinity caused by the expansion and π–π stacking of the $PEDOT^+$ chains[31]. In the high-resolution Ti 2$p$ XPS spectra of $Ti_3C_2T_x$ film and $Ti_3C_2T_x$ hydrogel (Fig. 2k), all $Ti_3C_2T_x$ hydrogel's peaks shift to higher binding energy by about 1.0 eV, together with a slightly increased Ti–C[32] contribution and somewhat decreased MXene surface groups (Supplementary Table 3). The peak corresponding to the C–Ti[33] bond also shifts from 281.8 eV ($Ti_3C_2T_x$ film) to 282.3 eV ($Ti_3C_2T_x$ hydrogel) in the C 1$s$ XPS spectra (Fig. 2l). Moreover, compared with $Ti_3C_2T_x$ film, the C–Ti bond contribution in $Ti_3C_2T_x$ hydrogel increases by three times (Supplementary Table 4), signifying the remarkably strengthened Ti–C interactions in the $Ti_3C_2T_x$ hydrogel. Considerable variation also occurs in the S 2$p$ XPS spectra, as depicted in Supplementary Fig. 14, where the ratio of $PSS^-$ to $PEDOT^+$ in pure PEDOT:PSS film is 2.2, while this value halves in $Ti_3C_2T_x$ hydrogel (1.1), a strong indication of the prominent removal of $PSS^-$ chains in $Ti_3C_2T_x$ hydrogel.

Generally, PEDOT:PSS tends to form a core-shell structure with the hydrophilic $PSS^-$ shell surrounding the hydrophobic $PEDOT^+$ core[34]. $PEDOT^+$ and $PSS^-$ chains interact via electrostatic attraction, and the electrostatic repulsion between ionic $PSS^-$ chains makes PEDOT:PSS hydrophilic and dispersible in water[35]. This feature enables the homogeneous blending of PEDOT:PSS with MXenes. Because of the electrostatic interaction and hydrogen bonding between MXene surface groups (−O, −OH, −F) and PEDOT:PSS[36], the original electrostatic interactions between $PEDOT^+$ and $PSS^-$ are altered. Partial $PEDOT^+$ chains attach to the negatively charged MXene surface and oxidize from benzene to quinoid configuration[37]. Once $H_2SO_4$ is introduced, the abundant $H^+$ ions further protonate $PSS^-$ and weaken the electrostatic attraction between $PEDOT^+$ and $PSS^-$ chains[38], as well as the repulsion force between PEDOT:PSS micelles[31]. As a result, more coiled $PEDOT^+$ chains expand to linear structures with elongated conjugation lengths, and more $PSS^-$ chains are removed from PEDOT:PSS and dissolved in the acidic solution. These exposed $PEDOT^+$ chains not only interact with each other to form physical crosslinks through π–π stacking and hydrophobic interaction[34] but also interact with MXene sheets via electrostatic attraction and hydrogen bonding and bridge them into 3D networks (Figs. 1, 2f–h and Supplementary Fig. 11). Meanwhile, the strengthened Ti–C bonds and reduced MXene surface groups are observed (Supplementary Tables 3 and 4), owing to the robust interactions between MXene and $PEDOT^+$. Polar DMSO is a secondary dopant[39] that promotes the removal of $PSS^-$ from PEDOT:PSS, like $H_2SO_4$, and they synergistically contribute to the self-assembly process (Supplementary Figs. 4 and 5). The heating treatment is indispensable during self-assembly, which accelerates these processes and strengthens the interactions, facilitating the generation of robust MXene hydrogels within a short time.

## High-rate electrochemical performance of 4D-printed MXene hydrogel electrodes

To demonstrate the feasibility of 4D-printed MXene hydrogels for electrochemical energy storage, we chose $Ti_3C_2T_x$ hydrogel as a model and investigated its electrochemical performance as supercapacitor electrode in Swagelok cell (Supplementary Fig. 15) (PEDOT:PSS hydrogel and $Ti_3C_2T_x$ film were also tested for comparison, Supplementary Fig. 16). As shown in Fig. 3a, $Ti_3C_2T_x$ hydrogel ($0.5\,mg\,cm^{-2}$) shows distinct redox peaks at approximately −0.3 V versus Ag/AgCl reference electrode at a scan rate of $10\,mV\,s^{-1}$, and negligible distortion of cyclic voltammetry (CV) curves is observed even at $10\,V\,s^{-1}$, indicating an ultrafast and highly reversible pseudocapacitive energy-storage process. This is also supported by the small cathodic and anodic peak potentials separation (less than 100 mV for scan rates up to $1\,V\,s^{-1}$) (Supplementary Fig. 17). In addition, differing from the loss of pseudocapacitive characteristics in the filtered $Ti_3C_2T_x$ electrode

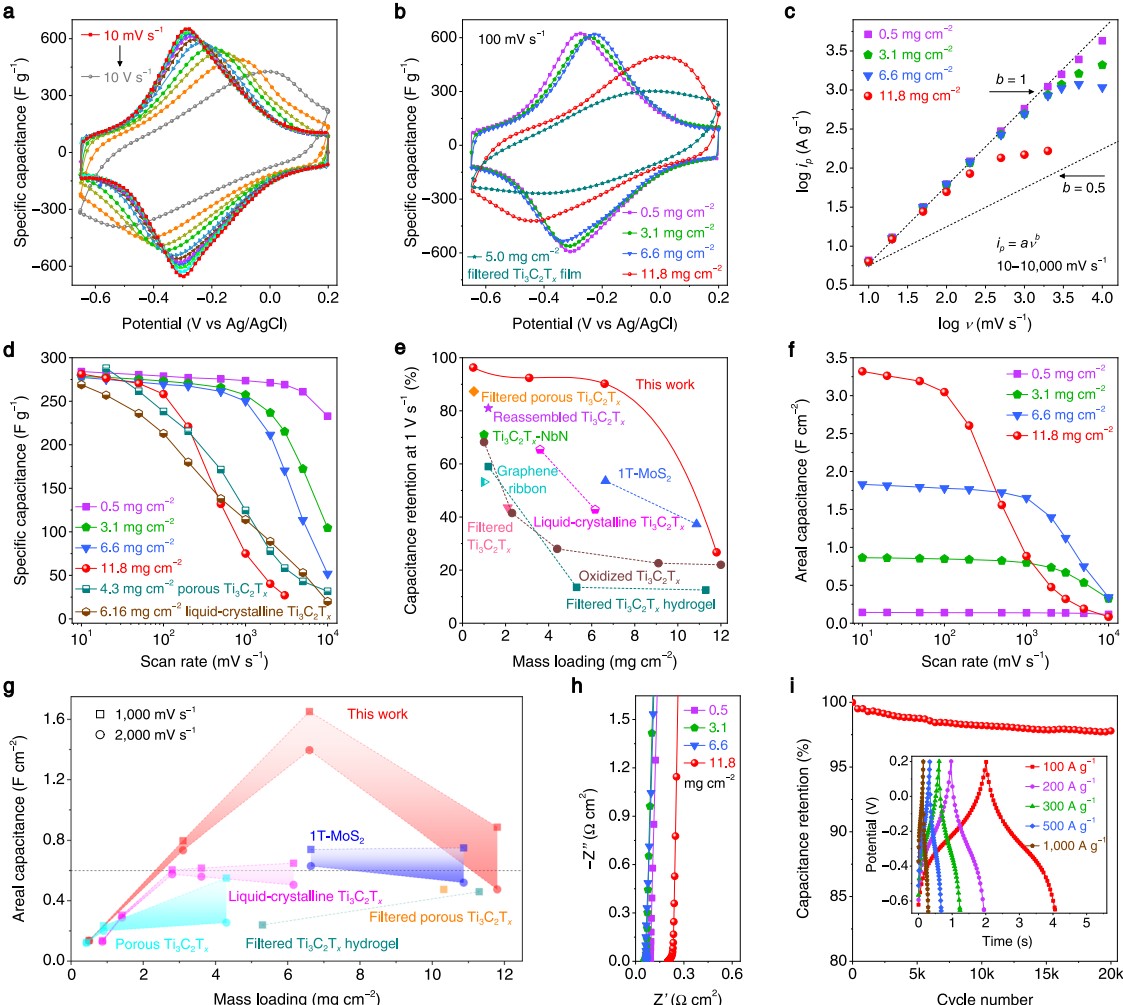

**Fig. 3 | Electrochemical performance of 4D-printed Ti₃C₂Tₓ hydrogel electrodes.** a CV curves of Ti₃C₂Tₓ hydrogel (0.5 mg cm⁻²) at scan rates of 10, 20, 50, 100, 200, 500, 1000, 2000, 3000, 5000, and 10,000 mV s⁻¹. **b** CV curves of Ti₃C₂Tₓ hydrogels with different mass loadings at a scan rate of 100 mV s⁻¹. **c** Determination of the slope, *b*-value, for Ti₃C₂Tₓ hydrogels with different mass loadings. **d** Rate performance of Ti₃C₂Tₓ hydrogels at scan rates from 10 to 10,000 mV s⁻¹. The porous Ti₃C₂Tₓ (4.3 mg cm⁻²)[12] and liquid-crystalline Ti₃C₂Tₓ (6.16 mg cm⁻²)[41] are plotted for comparison. **e** Capacitance retentions of Ti₃C₂Tₓ hydrogels with different mass loadings at 1 V s⁻¹. Lots of high-rate electrodes are listed for comparison, including filtered porous Ti₃C₂Tₓ[43], reassembled Ti₃C₂Tₓ[50], Ti₃C₂Tₓ-NbN[51], graphene ribbon[52], filtered Ti₃C₂Tₓ[53], liquid-crystalline Ti₃C₂Tₓ[41], 1T-MoS₂[42],

oxidized Ti₃C₂Tₓ[54], and filtered Ti₃C₂Tₓ hydrogel[12]. **f** Areal capacitance of Ti₃C₂Tₓ hydrogels with different mass loadings at scan rates from 10 to 10,000 mV s⁻¹. **g** Comparison of the areal capacitance of Ti₃C₂Tₓ hydrogels with benchmark electrodes at scan rates of 1 and 2 V s⁻¹. These electrodes are porous Ti₃C₂Tₓ[12], liquid-crystalline Ti₃C₂Tₓ[41], 1T-MoS₂[42], filtered Ti₃C₂Tₓ hydrogel[12], and filtered porous Ti₃C₂Tₓ[43]. **h** EIS plots of Ti₃C₂Tₓ hydrogels with different mass loadings taken at 0.2 V. **i** Long-term stability of Ti₃C₂Tₓ hydrogel performed by cycling at 100 mV s⁻¹. The inset depicts galvanostatic charge-discharge (GCD) profiles of Ti₃C₂Tₓ hydrogel (1.0 mg cm⁻²) at ultrahigh current densities of 100, 200, 300, 500, and 1000 A g⁻¹, respectively. The GCD curves of this Ti₃C₂Tₓ hydrogel at low current densities of 1, 2, 3, 5, and 10 A g⁻¹ are shown in Supplementary Fig. 20.

(5.0 mg cm⁻²) at a medium scan rate of 100 mV s⁻¹ (Fig. 3b), redox peaks of Ti₃C₂Tₓ hydrogels remain independent of mass loading/thickness from 0.5 mg cm⁻² (0.12 mm) to 6.6 mg cm⁻² (1.5 mm), and are evident even in a 2.9 mm-thick hydrogel with a commercially relevant mass loading of 11.8 mg cm⁻². This suggests the substantially enhanced ion-transport properties and mass loading/thickness-independent behavior of Ti₃C₂Tₓ hydrogels.

To interpret the capacitive performance of Ti₃C₂Tₓ hydrogels, charge-storage kinetic analysis was carried out (Fig. 3c). The peak-current density (*i*ₚ) and scan rate (*v*) from CV curves follow the relationship: $i_p = av^b$, where *a* and *b* are variables. *b*-value is an indicator of the charge-storage kinetics, which is determined from the slope of the plot of log *i*ₚ versus log *v*[40]. Apparently, the *b*-value remains close to 1 in a wide scan-rate range from 10 to 2000 mV s⁻¹ for Ti₃C₂Tₓ hydrogels with mass loadings up to 6.6 mg cm⁻², revealing that their charge-storage kinetics is surface-controlled. The diffusion-controlled mechanism becomes prominent when the mass loading of Ti₃C₂Tₓ

hydrogel reaches 11.8 mg cm⁻², which should be ascribed to the long ion diffusion distance in thick electrodes.

Figure 3d shows the rate performance of Ti₃C₂Tₓ hydrogels with different mass loadings/thicknesses. It is worth noting that, at an ultrahigh scan rate of 10 V s⁻¹, the 0.12 mm-thick hydrogel (0.5 mg cm⁻²) delivers a record high specific capacitance of 232.9 F g⁻¹, retaining 82% capacitance at 10 mV s⁻¹. For thicker and higher mass loading hydrogels (0.7 mm/3.1 mg cm⁻², 1.5 mm/6.6 mg cm⁻²), their rate performance decays are also minor at scan rates up to 1 V s⁻¹, superior to the previously reported 180 μm-thick porous Ti₃C₂Tₓ (4.3 mg cm⁻²)[12] and 320 μm-thick liquid-crystalline Ti₃C₂Tₓ (6.16 mg cm⁻²)[41]. The mass loading/thickness-independent rate performance of Ti₃C₂Tₓ hydrogels is further demonstrated in Fig. 3e. At a fixed high scan rate of 1 V s⁻¹, ultrahigh capacitance retentions of 96.3%, 92.4%, and 90.2% are achieved for Ti₃C₂Tₓ hydrogels with 0.5, 3.1, and 6.6 mg cm⁻² mass loadings, respectively, pronouncedly surpassing most state-of-the-art high-rate electrodes (Supplementary Table 5).

$Ti_3C_2T_x$ hydrogel electrodes also show remarkable areal capacitance. As displayed in Fig. 3f, the maximum areal capacitance of $Ti_3C_2T_x$ hydrogel (11.8 mg cm$^{-2}$) reaches 3.32 F cm$^{-2}$ at 10 mV s$^{-1}$, greater than most advanced MXene electrodes (Supplementary Table 5). Meanwhile, the areal capacitance grows linearly with mass loadings from 0.5 to 11.8 mg cm$^{-2}$ at scan rates up to 100 mV s$^{-1}$ (Supplementary Fig. 18). At a high scan rate of 2 V s$^{-1}$, this linear relationship is still maintained within a mass loading range of 0.5–6.6 mg cm$^{-2}$ (Fig. 3g), indicating fast ion diffusion and high-efficiency energy-storage in $Ti_3C_2T_x$ hydrogels. In commercial applications, the typical areal capacitance of electrode is 0.6 F cm$^{-2}$ (horizontal dash line in Fig. 3g)[42], and to date, only the liquid-crystalline $Ti_3C_2T_x$[41] and 1T-$MoS_2$[42] electrodes have achieved this value at 1 V s$^{-1}$. Hydrogel electrodes not only meet the commercial requirements but also possess two times higher areal capacitance than liquid-crystalline $Ti_3C_2T_x$ and 1T-$MoS_2$ electrodes at both 1 and 2 V s$^{-1}$ under similar mass loadings (6.16–6.6 mg cm$^{-2}$). At the maximum mass loading of 11.8 mg cm$^{-2}$, the areal capacitance of our $Ti_3C_2T_x$ hydrogel at 1 V s$^{-1}$ retains 0.89 F cm$^{-2}$, which is higher than 1T-$MoS_2$[42], filtered porous $Ti_3C_2T_x$[43], and filtered $Ti_3C_2T_x$ hydrogel[12] electrodes.

Electrochemical impedance spectroscopy (EIS) measurements were conducted to offer deeper insights into the ion transport and charge transfer behavior in MXene hydrogels. As displayed in Fig. 3h, all $Ti_3C_2T_x$ hydrogel electrodes, including the 2.9 mm-thick one (11.8 mg cm$^{-2}$), show low series resistances (less than 0.2 Ω cm$^2$). However, the ion diffusion resistance (Warburg impedance) between the MXene hydrogels and filtered MXene films is quite different (Supplementary Fig. 19). Unlike the filtered $Ti_3C_2T_x$ electrode with a clear 45° slope in the mid-frequency region, plots of all $Ti_3C_2T_x$ hydrogels are nearly vertical at all the measured frequencies, certainly demonstrating the faster ion diffusion in hydrogel electrodes, which is vital for mass loading/thickness-independent performance. Moreover, the $Ti_3C_2T_x$ hydrogel is highly stable, remaining 97.8% capacitance after cycling for 20,000 times at a scan rate of 100 mV s$^{-1}$ (Fig. 3i). All these results show a potential of MXene hydrogels to enable ultrahigh charge-discharge rates and long-term cycling stability within thick electrodes with high mass loading.

**Charge-storage performance of 4D-printed MSCs**
To investigate the suitability of MXene hydrogels for practical energy-storage applications, a symmetric $Ti_3C_2T_x$ hydrogel MSC (mass loading ~35 mg cm$^{-2}$) was 4D-printed and tested in a polyvinyl alcohol-ethylene glycol-sulfuric acid (PVA-EG-$H_2SO_4$) gel electrolyte (Supplementary Fig. 21). Notably, the highly conductive $Ti_3C_2T_x$ hydrogel (1548 S m$^{-1}$) does not require current collector. The CV curves of the 4D-printed $Ti_3C_2T_x$ hydrogel MSC at scan rates from 2 to 100 mV s$^{-1}$ at room temperature (25 °C) are shown in Fig. 4a. The stable operating voltage window of this MSC reaches 0.6 V, together with a pair of redox peaks observed at around 0 V, which is an indicator of the pseudocapacitive behavior and agrees with the nonlinear GCD curves (Fig. 4b). According to the CV data, the areal capacitance of the 4D-printed $Ti_3C_2T_x$ hydrogel MSC reaches 2.31 F cm$^{-2}$ at a scan rate of 2 mV s$^{-1}$ and retains 0.53 F cm$^{-2}$ when the scan rate is increased by 50-fold (100 mV s$^{-1}$) (Supplementary Fig. 22). This performance surpasses most of printed MSCs (Fig. 4c, Supplementary Table 6), such as 3D-printed $Ti_3C_2T_x$ MSC (2.1 F cm$^{-2}$ at 1.7 mA cm$^{-2}$)[26], 3D-printed graphene MSC (1.57 F cm$^{-2}$ at 2 mA cm$^{-2}$)[44], and screen-printed $Ti_3C_2T_x$ MSC (0.158 F cm$^{-2}$ at 0.08 mA cm$^{-2}$)[45]. The areal energy and power densities of the 4D-printed $Ti_3C_2T_x$ hydrogel MSC are also extraordinary. As displayed in Fig. 4d, the maximum areal energy and power densities of our MSC achieve 92.88 µWh cm$^{-2}$ and 6.96 mW cm$^{-2}$, respectively. These values are greater than most reported MSCs (Supplementary Table 7), for instance, 3D-printed $Ti_3C_2T_x$ MSCs (51.7 µWh cm$^{-2}$, 4 mW cm$^{-2}$)[46], screen-printed $Ti_3C_2T_x$

MSC (1.64 µWh cm$^{-2}$, 0.778 mW cm$^{-2}$)[45], and direct-written graphene-CNT MSC (1.36 µWh cm$^{-2}$, 0.25 mW cm$^{-2}$)[47]. It is worth mentioning that, upon optimizing the configuration of the devices (e.g., mass loading and finger electrode gap), the electrochemical performance of our MSC could be further enhanced[45].

Low-temperature adaptability is crucial for electrochemical energy-storage devices in practical applications, but accomplishing this goal remains challenging due to the reduced mobility of electrolyte near/below their freezing points[48,49]. Importantly, the PVA-EG-$H_2SO_4$ gel electrolyte we used is not frozen but shows great transparency and mechanical flexibility even at −20 °C (Fig. 4e inset), differing from the icy PVA-$H_2SO_4$ electrolyte with a white color (Supplementary Fig. 23). This can be attributed to the anti-freezing feature of EG towards water and the abundant hydrogen bonds between EG molecules and PVA chains[49]. More importantly, the ionic conductivity of PVA-EG-$H_2SO_4$ maintains as high as 27.5 mS cm$^{-1}$ at −20 °C (Supplementary Fig. 24, Supplementary Table 8), which facilitates the rapid electrolyte ions diffusion at subzero temperatures. The highly conductive and porous $Ti_3C_2T_x$ hydrogel with enhanced electrolyte accessibility also potentially alleviates the adverse effects of low temperatures. Consequently, excellent low-temperature tolerance is achieved for the 4D-printed $Ti_3C_2T_x$ hydrogel MSC, retaining 90.6% capacitance at 0 °C and 82.2% capacitance at −20 °C, in comparison with that at 25 °C (Fig. 4e). These values are significantly higher than other reports (Supplementary Fig. 25). Even after cooling and heating for several cycles between −20, 0, and 25 °C, the capacitance of our MSC recovers, demonstrating its remarkable low-temperature reversibility (Fig. 4f). Moreover, at −20 °C, the areal capacitance of 4D-printed $Ti_3C_2T_x$ hydrogel MSC reaches 1.42 F cm$^{-2}$ (10 mV s$^{-1}$), superior to most high-performance low-temperature tolerant supercapacitors ever reported (Fig. 4g) (Supplementary Table 9).

To further demonstrate the potential of 4D-printed $Ti_3C_2T_x$ hydrogel MSC for practical application, particularly in cold environments, a cycling test at −20 °C was conducted. As shown in Fig. 4h, our MSC retains ~81% of initial capacitance and ~100% Coulombic efficiency after 10,000 cycles. The excellent electrochemical durability is also demonstrated by the nearly unaltered EIS curves before and after cycling (Fig. 4h insert). In addition, our MSC units can be arbitrarily connected in series and/or in parallel to meet the specific energy/power requirements in real scenarios (Fig. 4i). For instance, a four in series (4 S) tandem device can readily power three LED bulbs (Fig. 4i insert).

## Discussion
We have developed an advanced 4D printing technology for the efficient fabrication of MXene hydrogels. This strategy is universal in its application to a family of MXenes with different atomic layers and transition metal types (e.g., $Nb_2CT_x$, $Ti_3C_2T_x$, and $Mo_2Ti_2C_3T_x$) and a series of MXene hydrogels with complex and precise architectures on various substrates were successfully manufactured. These included a $Nb_2CT_x$ hydrogel Chinese knot on cloth, a $Nb_2CT_x$ hydrogel "CRANN" logo on PET film, a $Ti_3C_2T_x$ hydrogel microlattice and a $Ti_3C_2T_x$ hydrogel rectangular hollow prism on glass slides, and $Mo_2Ti_2C_3T_x$ hydrogel micro-supercapacitor units on PET film. The obtained MXene hydrogels all present 3D porous structures, large specific surface areas, and high electrical conductivities. As a result, highly efficient pseudocapacitive energy storage could be achieved, including ultrahigh capacitances, excellent mass loading/thickness-independent rate capabilities, great low-temperature tolerances, and high areal energy/power densities (92.88 µWh cm$^{-2}$, 6.96 mW cm$^{-2}$). This work offers new insights into the manufacture of MXene hydrogels and will advance the applications of MXenes and conductive hydrogels in electrochemical energy storage and conversion, sensors, bioelectronics, electromagnetic interference shielding, water purification, and other technologies.

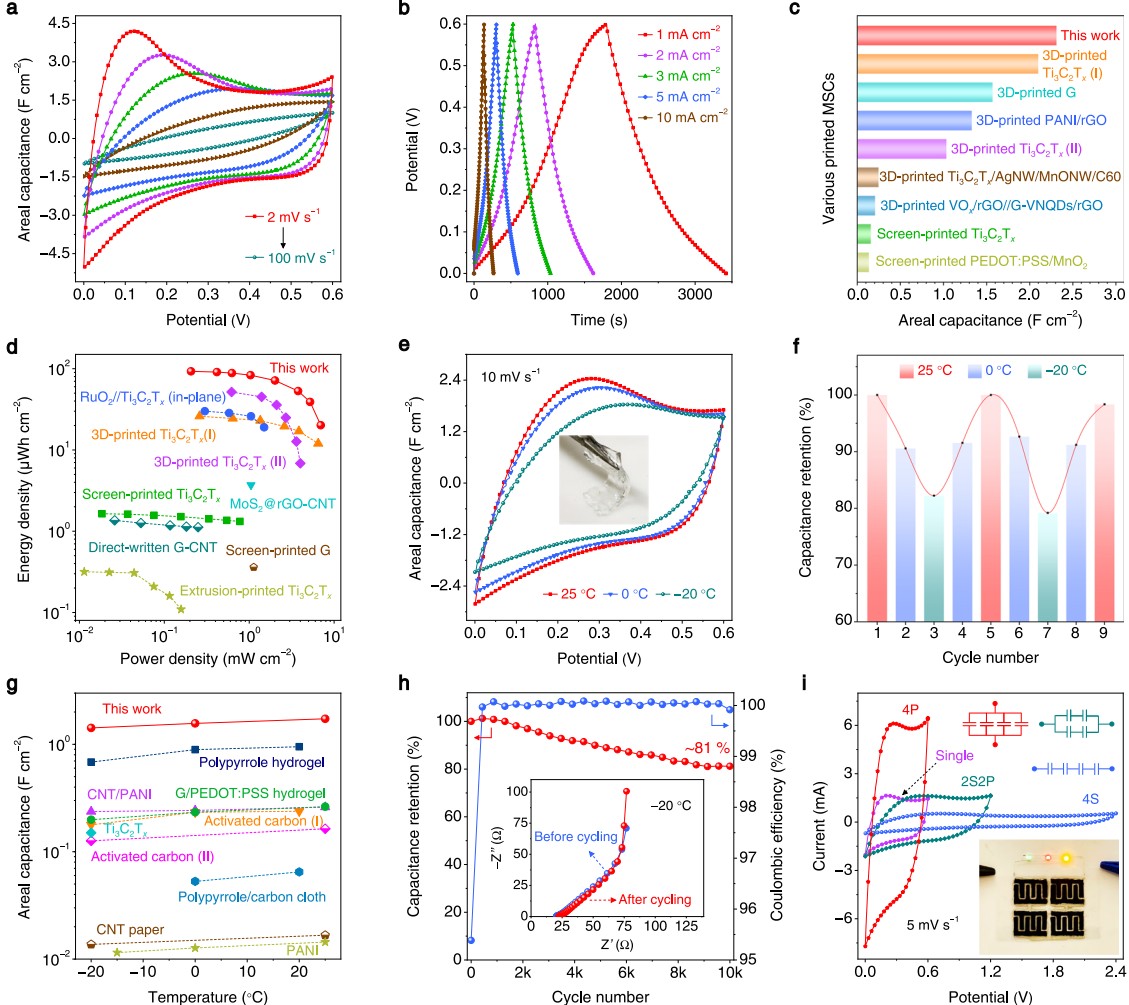

**Fig. 4 | Electrochemical performance of 4D-printed Ti$_3$C$_2$T$_x$ hydrogel MSCs.**
**a** CV curves of 4D-printed Ti$_3$C$_2$T$_x$ hydrogel MSC at scan rates of 2, 5, 10, 20, 50, and 100 mV s$^{-1}$. **b** GCD curves of 4D-printed Ti$_3$C$_2$T$_x$ hydrogel MSC at current densities of 1, 2, 3, 5, and 10 mA cm$^{-2}$. **c** Comparison of areal capacitance of 4D-printed Ti$_3$C$_2$T$_x$ hydrogel MSC with other printed MSCs. **d** Ragone plots of 4D-printed Ti$_3$C$_2$T$_x$ hydrogel MSC and other high-performance MSCs. **e** CV curves of 4D-printed Ti$_3$C$_2$T$_x$ hydrogel MSC at a scan rate of 10 mV s$^{-1}$ at 25 °C, 0, and −20 °C. Inset is the PVA·EG·H$_2$SO$_4$ gel electrolyte at −20 °C, showing great transparency and flexibility. **f** Capacitance retention of 4D-printed Ti$_3$C$_2$T$_x$ hydrogel MSC during cooling/heating cycles. **g** Comparison of areal capacitance of 4D-printed Ti$_3$C$_2$T$_x$ hydrogel MSC (10 mV s$^{-1}$) with other devices at low temperatures. **h** Cycling performance of 4D-printed Ti$_3$C$_2$T$_x$ hydrogel MSC at a current density of 30 mA cm$^{-1}$ at −20 °C, inset shows the EIS data of this MSC before and after 10,000 cycles at −20 °C. **i** CV curves of single 4D-printed Ti$_3$C$_2$T$_x$ hydrogel MSC, four MSCs in series (4 S), four MSCs in parallel (4 P), and two in series and in parallel (2S2P), at a scan rate of 5 mV s$^{-1}$. Insert shows the photograph of a 4 S tandem device powering three LED indicators, demonstrating the feasibility of our MSCs for practical applications.

## Methods

### Synthesis of Mo$_2$Ti$_2$AlC$_3$ MAX phase

To synthesize the Mo$_2$Ti$_2$AlC$_3$ MAX precursor, Mo (250 mesh, Alfa Aesar, 99.9%), Ti (325 mesh, Alfa Aesar, 99.5%), Al (325 mesh, Alfa Aesar, 99.5%), and graphite (325 mesh, Alfa Aesar, 99%) powders were used, with an atomic ratio of 2:2:1.3:2.7 (50 g total). The powder mixtures were then mixed in a 2:1 ball:powder ratio with 5 mm alumina balls, and ball milled at 60 rpm for 24 h. Afterward, high-temperature sintering reactions at 1600 °C for 4 h in a Carbolite furnace were conducted, with a heating and cooling rate of 3 °C min$^{-1}$, and 200 cm$^3$ min$^{-1}$ flow of ultrahigh purity Ar (99.999%). After cooling down, the porous compact product was milled using a TiN-coated milling bit and sieved through a 400-mesh sieve, producing powders with a particle size <38 μm. The powders were then added into 9 M HCl (Sigma–Aldrich, 37%) to dissolve any residual metals or intermetallics and washed with deionized water by filtration until neutral. All experiments on this study were conducted on a single batch of MAX to eliminate any artifacts from variation between MAX synthesis batches.

### Synthesis of few-layer Nb$_2$CT$_x$

One gram of Nb$_2$AlC MAX (400 mesh, 98%, Jilin 11 Technology, Co. Ltd, China) was first added slowly (over 10 min) into a mixture solution consisting of 2 g of LiF (Alfa Aesar, 97%) and 20 mL of 12 M HCl and stirred at 60 °C for 96 h, yielding multilayer Nb$_2$CT$_x$, which was further washed with 1 M HCl and deionized water by centrifugation (Heraeus Multifuge X1 Centrifuge, Thermo Fisher Scientific, USA) at 3005 × g (4000 rpm) until the supernatant became neutral. To produce few-layer Nb$_2$CT$_x$, the as-synthesized multilayer Nb$_2$CT$_x$ was dispersed in 20 mL of 5 wt. % (diluted) tetramethylammonium hydroxide (TMAOH) solution (Sigma–Aldrich, 25 wt.%) and stirred at room temperature for 12 h. Afterward, the mixture was centrifuged directly at 4696 × g (5000 rpm) for 20 min, and then washed with 40 mL of deionized water by centrifugation at 15,777 × g (12,000 rpm) to remove excess TMAOH. The obtained precipitate was redispersed in deionized water and sonicated for 1 h in an ice bath with N$_2$ bubbling. After another centrifugation at 3005 × g (4000 rpm) for 1 h, few-layer Nb$_2$CT$_x$ dispersion with a dark color was collected.

## Synthesis of few-layer $Ti_3C_2T_x$

One gram of $Ti_3AlC_2$ MAX (400 mesh, Carbon, Ukraine) was first added slowly (over 10 min) into a mixture solution consisting of 1 g of LiF and 20 mL of 9 M HCl and stirred at 35 °C for 48 h, yielding multilayer $Ti_3C_2T_x$, which as further washed with deionized water by centrifugation at $3005 \times g$ until the supernatant became neutral. To produce few-layer $Ti_3C_2T_x$, the as-synthesized multilayer $Ti_3C_2T_x$ was redispersed in deionized water and sonicated for 1 h in an ice bath with $N_2$ bubbling. After another centrifugation at $3005 \times g$ for 1 h, few-layer $Ti_3C_2T_x$ dispersion with a dark color was collected.

## Synthesis of few-layer $Mo_2Ti_2C_3T_x$

One gram of $Mo_2Ti_2AlC_3$ MAX was first added slowly (over 10 min) into 20 mL of HF (Acros Organics, 48–50 wt.%) and stirred at 55 °C for 96 h, yielding multilayer $Mo_2Ti_2C_3T_x$, which was further washed with deionized water by centrifugation at $3005 \times g$ until the supernatant became neutral. To produce few-layer $Mo_2Ti_2C_3T_x$, the as-synthesized multilayer $Mo_2Ti_2C_3T_x$ was dispersed in 20 mL of 5 wt. % TMAOH solution and stirred at 35 °C for 12 h. Afterward, the mixture was centrifuged directly at $4696 \times g$ for 20 min, and then washed with 40 mL of deionized water by centrifugation at $15,777 \times g$ to remove excess TMAOH. The obtained precipitate was redispersed in deionized water and sonicated for 1 h in an ice bath with $N_2$ bubbling. After another centrifugation at $3005 \times g$ for 1 h, few-layer $Mo_2Ti_2C_3T_x$ dispersion with a dark color was collected.

## Preparation of MXene hydrogels by self-assembly method in molds

Typically, 3.2 mL of few-layer $Ti_3C_2T_x$ MXene suspension (20 mg mL$^{-1}$) and 1.52 mL of PEDOT:PSS suspension (PH 1000, Clevios$^{TM}$, ~10.5 mg mL$^{-1}$) were first mixed into a homogenous dispersion by stirring for 30 min and sonication for 10 min. Another mixture solution containing 2.6 mL of DMSO (Sigma–Aldrich, 99.9%, anhydrous), 0.33 mL of 3 M $H_2SO_4$ (Honeywell, 95–97%), 1.15 mL of 1 M sodium L-ascorbate (Acros Organics, 99%), and 1.2 mL of deionized water was added in dropwise and kept stirring for 30 min. Afterward, the obtained black mixture with a concentration of 8 mg mL$^{-1}$ was transferred into sealed molds (e.g., glass vials and capillary tubes) with different shapes and sizes and heated at 90 °C for 6 h in oven, resulting in black $Ti_3C_2T_x$ MXene hydrogels (MXene mass content: 80 wt.%). The as-prepared hydrogels were further immersed in 3 M and 18.4 M $H_2SO_4$ for 24 h to enhance their mechanical strength and washed with deionized water to remove any impurities. Notably, when optimizing the self-assembly conditions, the mass content of MXene in MXene-PEDOT:PSS mixture varied from 0, 30, 50, 70, 80, 90, to 100 wt.%, the volume proportion of DMSO to the whole dispersion (include MXene suspension, PEDOT:PSS suspension, additives and deionized water) varied from 0, 13, 26, to 50 vol.%, and the concentration of $H_2SO_4$ in the whole dispersion was set as 0.1 M or 1 M. The molar ratio of the reducing agent sodium L-ascorbate to metal atom in MXene was fixed at 1:1.

## 4D printing of MXene hydrogels

A black mixture dispersion of MXene ($Nb_2CT_x$, $Ti_3C_2T_x$, or $Mo_2Ti_2C_3T_x$), PEDOT:PSS, and additives (0.1 M $H_2SO_4$, 26 vol.% DMSO and specific amount sodium L-ascorbate) was first condensed by centrifugation at $10,956 \times g$ (10,000 rpm), resulting in solid-like inks with MXene-PEDOT:PSS presenting a concentration of ~50 mg mL$^{-1}$. The inks were then loaded into a 5 mL syringe with a stainless-steel nozzle (0.26 mm) and being 3D-printed into different architectures via a commercial 3D printer (nano3Dprint). The printing speed was set as 1000 mm min$^{-1}$. After 3D printing, the obtained MXene sols were sealed in a glass container and heated at 90 °C for 6 h in oven, to accomplish the transformation of MXene sols into MXene hydrogels. These MXene hydrogels were further immersed in 3 M and 18.4 M $H_2SO_4$ for 24 h to enhance their mechanical strength (3 M and 12 M $H_2SO_4$ employed when PET substrates were utilized to avoid their dissolution). Afterward, the obtained MXene hydrogels were immersed in deionized water and washed several times to remove any impurities. The mass contents of MXenes in three hydrogels are 80 wt.% and the solid contents of three MXene hydrogels are ~4.2 wt.%.

## Preparation of low-temperature tolerant polymer gel electrolyte

2.67 mL of EG (Hach) was first mixed with 5.33 mL of deionized water, then 1 g of PVA (Aldrich, Mw 89,000–98,000, 99%, hydrolyzed) was introduced and stirred at 85 °C for 4 h until dissolved. After cooling down completely, another solution consisting of 3 g of 18.4 M $H_2SO_4$, 0.67 mL of EG, and 1.33 mL of deionized water was added in slowly, and stirred for another hour, yielding the PVA-EG-$H_2SO_4$ gel electrolyte.

## Fabrication of micro-supercapacitors

4D-printed MXene hydrogel MSC (Supplementary Fig. 21, mass loading ~35 mg cm$^{-2}$, size $2.2 \times 1.7 \times 0.4$ cm (L × W × H)) were vacuum-dried (~0.5 mbar) at room temperature first, then two silver wires were connected separately to the two electrodes by conductive silver paint. After completely drying, nail polish was further coated on the top of Ag paint to protect it from potential exposure to electrolytes. Afterward, a layer of PVA-EG-$H_2SO_4$ gel electrolyte (1.8 mL) was cast, and the obtained MSCs were stored in a vacuum desiccator for 10 min to facilitate the electrolyte permeation. It is worth noting that, benefiting from the high electrical conductivity of MXene hydrogels, no extra metal current collectors were employed. The size of devices shown in Fig. 4i is $2.2 \times 1.7 \times 0.05$ cm, and 0.22 mL of gel electrolyte was employed.

## Material characterization

The morphology of MXene hydrogels was observed using SEM (Zeiss ULTRA plus) and TEM (FEI Titan 80–300, 300 kV). Surface chemical information of the samples was analyzed by an Omicron Multiprobe XPS instrument equipped with a monochromatic Al Kα X-ray source (h$\nu$ = 1486.6 eV). Raman spectra were recorded with a Horiba Labram Aramis Raman spectrometer at 633 nm (grating 1800 gr/mm, 10% of laser power). XRD patterns were obtained using a powder diffractometer (Bruker D8 Discover) with Cu Kα1 radiation. The $N_2$ adsorption/desorption measurements were performed on Quantachrome Autosorb-iQ. Before measurement, the samples (vacuum-dried at room temperature) were further vacuum-dried at 150 °C for 10 h. The rheological properties of the inks were investigated using an Anton Paar MCR 302e rheometer using a 50 mm plate-plate geometry. The upper plate was a roughened plate (PP50/S) to lessen the effects of slip and a solvent trap was used to minimize the effects of evaporation.

## Electrochemical measurements

All electrochemical tests were performed on a Biologic VMP300 potentiostat. For the three-electrode test, plastic Swagelok cells (PFA-420-3) were used (Supplementary Fig. 15). MXene hydrogels on glassy carbon, overcapacitive activated carbon (YP-50, Kuraray, Japan), Ag/AgCl in 3.5 M KCl, and 3 M $H_2SO_4$ (0.5 mL) were used as working electrode, counter electrode, reference electrode, and electrolyte respectively. For the MSC device test, vacuum-dried MXene hydrogels and PVA-EG-$H_2SO_4$ were the electrode materials and gel electrolyte, respectively. The room-temperature electrochemical tests were done in ambient conditions with an average temperature of 25 °C. The low-temperature electrochemical tests were conducted in a lab fridge. EIS was measured ranging from 10 mHz to 1000 kHz under a potential amplitude of 10 mV.

## Calculations for the electrochemical tests

$b$-value calculation: the highest current $i$ during discharging, or the current at the highest peak, was recorded as a function of the scan rate $v$ in the range of 10–10,000 mV s$^{-1}$. By fitting log $i$ as a function of log $v$, a linear curve was obtained with its slope as the $b$ value.

The specific capacitance (F g$^{-1}$) of a single electrode was calculated based on CV curves:

$$C_{sp} = \frac{\int i \cdot t}{m \times V},$$ (1)

where $i$ is the current, $t$ is the discharging time, $m$ is the mass of entire electrode, $V$ is the voltage window of CV scan.

The areal capacitance (F cm$^{-2}$) of a single electrode was calculated as:

$$C_s = M_e \times C_{sp},$$ (2)

where $M_e$ is the mass loading of entire electrode.

As the GCD curves of electrodes are not linear, the capacity (mAh g$^{-1}$) of a single electrode, instead of capacitance, was calculated from the GCD curve as:

$$C = \frac{i \times \triangle t}{3.6 \times m},$$ (3)

where $i$ is the current, $\triangle t$ is the discharging time, and $m$ is the mass of entire electrode.

For the two-electrode device (MSC), the areal capacitance (F cm$^{-2}$) was evaluated based on CV curves:

$$C_{s,\text{MSC}} = \frac{\int i \cdot t}{S_{\text{MSC}} \times V_{\text{MSC}}},$$ (4)

where $i$ is the current, $t$ is the discharging time, $S_{MSC}$ is the entire area of MSC (including both the electrodes and gap), $V_{MSC}$ is the voltage window of MSC.

The areal energy density (mWh cm$^{-2}$) and average power density (mW cm$^{-2}$) of MSC was calculated based on the discharge scan of GCD curves:

$$E_{\text{MSC}} = \frac{i}{S_{\text{MSC}}} \times \left( \int_0^{t_{\text{MAX}}} V_{\text{MSC}} \cdot dt \right),$$ (5)

and

$$P_{\text{MSC}} = \frac{E_{\text{MSC}}}{\triangle t},$$ (6)

where $i$ is the current, $S_{MSC}$ is the entire area of MSC (including both the electrodes and gap), $t_{MAX}$ is the maximum discharge time, $V_{MSC}$ is the voltage window of MSC, $\triangle t$ is the discharging time.

Ionic conductivity (mS cm$^{-1}$) of PVA-EG-H$_2$SO$_4$ gel electrolyte was calculated according to this formula:

$$\sigma = \frac{L}{R \times S},$$ (7)

where $L$ is the thickness of electrolyte, $R$ is the resistance of electrolyte, and $S$ is the cross-section area of electrolyte.

## Data availability

The data that support the findings of this study are available from the corresponding authors upon reasonable request. Source data are provided with this paper.

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

## Acknowledgements

We acknowledge the Advanced Microscopy Laboratory for the provision of their facilities. This research was sponsored by the European Research Council ((CoG 3D2Dprint (GA 681544) and PoC Powering_eTextiles (GA 861673)) and Science Foundation Ireland (12/RC/2278_P2 and 16/RC/3872). L.H. thanks the Irish Research Council (GOIPG/2019/2642). A.Z. thanks the Irish Research Council (GOIPD/2022/443). I.V.S. thanks the Irish Research Council (IRCLA/2019/171). M.M. thanks the Science Foundation Ireland (17/CDA/4704). MXene development at Drexel University was supported by the US National Science Foundation grant DMR-2041050. For the purpose of Open Access, the author has applied a CC BY public copyright license to any Author Accepted Manuscript version arising from this submission.

## Author contributions

K.L., Y.G., and V.N. conceived the project. K.L., J.Z., and C.E.S. synthesized the MXenes. K.L., J.Z., and Y.D. conducted the 3D printing and MXene hydrogels preparation. K.L. performed SEM, Raman, and XRD analyses. A.Z. and I.V.S. performed XPS measurements. L.H. performed TEM analysis. S.B. and M.M. performed rheology analysis. S.V. and W.S. performed BET analysis. K.L. performed electrochemical tests and data analysis. K.L. wrote the manuscript, all authors discussed the results and commented on the manuscript.

## Competing interests

The authors declare no competing interests.
