## [Peer Review File · Nature Communications]

Reviewer comments, first round review

Reviewer #1 (Remarks to the Author):

The authors report a 4D printing approach for manufacturing MXene hydrogels. This strategy shows remarkable universality, which suits not only the most commonly studied Ti₃C₂T_x MXene but also Nb₂C_{Tx} and Mo₂Ti₂C₃T_x. These three MXenes with different atomical structures and properties can represent a family of MXenes. It has never been achieved in previous reports. Besides, the printed MXene hydrogel architectures are precise and customizable, which is unlikely to realize in traditional model-dependent methods. The formation mechanism of MXene hydrogel is also clearly investigated. In capacitive application, the thickness/mass loading-independent rate capability of MXene hydrogel is very impressive, which surpasses the most advanced electrodes ever reported. The areal capacitance of MXene hydrogel also meets the commercial requirements even at a high scan rate of 1 V s⁻¹. The further printed micro-supercapacitor delivers outstanding capacitive performance and prominent low-temperature tolerance. This work is very interesting, which will advance the development of both MXenes and hydrogels and attract a wide range of attention in different fields, including additive manufacturing, material science, bioengineering, electrochemistry, etc. It is recommended to be published on Nature Communications after minor revision. Followings are some comments:

1. The electrical conductivity of Ti₃C₂T_x hydrogel is much higher than the other two MXene hydrogels, the authors should discuss the possible reasons, and present the electrical conductivity values of three MXene films for comparison. In addition, electrical conductivity is crucial in various applications, such as biomedical, sensing, and electrochemistry. The comparison of the electrical conductivity of the MXene hydrogels with other reports should be presented, as well as their application fields. This will reveal the other potential applications of MXene hydrogels.
2. How about the capacitive performance of pure PETOT:PSS hydrogel and Ti₃C₂T_x MXene film?
3. In the Introduction section (Line 58), the authors claim that the 4D printing technology is scalable, but only several architectures are shown in this manuscript. The authors should manufacture more hydrogel units to demonstrate this property.
4. Five kinds of architectures are shown in Figure 2e, but only 2 corresponding videos are presented in Supplementary Movies. It would be better to show all the relating videos.
5. The micro-supercapacitors exhibit excellent low-temperature tolerance and the PVE-EG-H₂SO₄ electrolyte is a key contributor. How about the ionic conductivity of this electrolyte in comparison with other similar electrolytes? This will advise the future development of low-temperature devices.
6. Some minor errors are found, such as "10 mins" (Line 399) and "20 min" (Line 405). Please carefully check the manuscript.

Reviewer #2 (Remarks to the Author):

In this manuscript, the author shows out a universal 4D printing technology for manufacturing MXene HGs with customizable geometries and obtain an excellent electrochemical performance in supercapacitors. Besides, the low-temperature performance of micro-supercapacitors is also tested. But in my opinion, this manuscript has the defects: (1) the novelty is not enough for publication in Nature Commination; (2) compared to 3D printing technology or other methods, the advantages of 4D printing technology are not highlighted. Therefore, I think that this manuscript should be major revised or published others. Following are some comments that are concerned:

1. Compared to 3D printing technology, what are the advantages of MXene HGs by 4D printing? The authors should give more discussion in the introduction.
2. The micro-supercapacitor delivers ultrahigh energy and power densities up to 93 μWh cm⁻² and 7 mW cm⁻², but compared to the previous literature, the power density value of this work cannot reach an ultrahigh value (e.g. Adv. Energy Mater.2020, 10, 1903794; Energy Environ. Sci., 2019, 12, 96--115).

3. In the "probing self-assembly mechanism" section, the authors mentioned that "suggesting the conformation change from benzene structure to quinoid structure and thus elongated conjugation lengths of PEDOT+ chains." The authors are suggested to add necessary citation to rationalize the discussion.
4. The electrical conductivity of Ti₃C₂T_x HGs can reach 1548 S m⁻¹, which is far more than the cases of Nb₂C_{Tx} and Mo₂Ti₂C₃T_x HGs. The authors should give detailed reasons in the manuscript.
5. More experimental details could be included in the manuscript. For instance, the electrolyte amount used in the devices should be provided, which is an important parameter to evaluate the electrochemical performance of supercapacitors. And the thickness of electrodes should be provided in the Methods section.
6. In Figure 2b, the green line cannot be found, the authors should clearly check the figures.
7. In Figure 3h, the scale range of x-axis and y-axis is inconsistent. In addition, have the EIS data been normalized by mass or surface area of electrode?
8. In page 13 of main text and Figure 4d, the author mentioned the energy density and power densities. The reviewer wonders that are these values calculated based on total mass of cells or only of electrode.

Reviewer #1 (Remarks to the Author):

The authors report a 4D printing approach for manufacturing MXene hydrogels. This strategy shows remarkable universality, which suits not only the most commonly studied Ti_3C_2Tx MXene but also Nb_2CTx and $Mo_2Ti_2C_3Tx$. These three MXenes with different atomical structures and properties can represent a family of MXenes. It has never been achieved in previous reports. Besides, the printed MXene hydrogel architectures are precise and customizable, which is unlikely to realize in traditional model-dependent methods. The formation mechanism of MXene hydrogel is also clearly investigated. In capacitive application, the thickness/mass loading-independent rate capability of MXene hydrogel is very impressive, which surpasses the most advanced electrodes ever reported. The areal capacitance of MXene hydrogel also meets the commercial requirements even at a high scan rate of 1 V s^{-1} . The further printed micro-supercapacitor delivers outstanding capacitive performance and prominent low-temperature tolerance. This work is very interesting, which will advance the development of both MXenes and hydrogels and attract a wide range of attention in different fields, including additive manufacturing, material science, bioengineering, electrochemistry, etc. It is recommended to be published on Nature Communications after minor revision. Followings are some comments:

1. The electrical conductivity of Ti_3C_2Tx hydrogel is much higher than the other two MXene hydrogels, the authors should discuss the possible reasons, and present the electrical conductivity values of three MXene films for comparison. In addition, electrical conductivity is crucial in various applications, such as biomedical, sensing, and electrochemistry. The comparison of the electrical conductivity of the MXene hydrogels with other reports should be presented, as well as their application fields. This will reveal the other potential applications of MXene hydrogels.
2. How about the capacitive performance of pure PETOT:PSS hydrogel and Ti_3C_2Tx MXene film?
3. In the Introduction section (Line 58), the authors claim that the 4D printing technology is scalable, but only several architectures are shown in this manuscript. The authors should manufacture more hydrogel units to demonstrate this property.
4. Five kinds of architectures are shown in Figure 2e, but only 2 corresponding videos are presented in Supplementary Movies. It would be better to show all the relating videos.
5. The micro-supercapacitors exhibit excellent low-temperature tolerance and the PVE-EG- H_2SO_4 electrolyte is a key contributor. How about the ionic conductivity of this electrolyte in comparison with other similar electrolytes? This will advise the future development of low-temperature devices.
6. Some minor errors are found, such as "10 mins" (Line 399) and "20 min" (Line 405). Please carefully check the manuscript.

Reviewer #2 (Remarks to the Author):

In this manuscript, the author shows out a universal 4D printing technology for manufacturing MXene HGs with customizable geometries and obtain an excellent electrochemical performance in supercapacitors. Besides, the low-temperature performance of micro-supercapacitors is also tested. But in my opinion, this manuscript has the defects: (1) the novelty is not enough for publication in Nature Commination; (2) compared to 3D printing technology or other methods, the advantages of 4D printing technology are not highlighted. Therefore, I think that this manuscript should be major revised or published others. Following are some comments that are concerned:

1. Compared to 3D printing technology, what are the advantages of MXene HGs by 4D printing? The authors should give more discussion in the introduction.

2. *The micro-supercapacitor delivers ultrahigh energy and power densities up to 93 $\mu\text{Wh cm}^{-2}$ and 7 mW cm^{-2} , but compared to the previous literature, the power density value of this work cannot reach an ultrahigh value (e.g. Adv. Energy Mater.2020, 10, 1903794; Energy Environ. Sci., 2019, 12, 96--115).*
3. *In the "probing self-assembly mechanism" section, the authors mentioned that "suggesting the conformation change from benzene structure to quinoid structure and thus elongated conjugation lengths of PEDOT+ chains." The authors are suggested to add necessary citation to rationalize the discussion.*
4. *The electrical conductivity of $\text{Ti}_3\text{C}_2\text{T}_x$ HGs can reach 1548 S m^{-1} , which is far more than the cases of Nb_2CT_x and $\text{Mo}_2\text{Ti}_2\text{C}_3\text{T}_x$ HGs. The authors should give detailed reasons in the manuscript.*
5. *More experimental details could be included in the manuscript. For instance, the electrolyte amount used in the devices should be provided, which is an important parameter to evaluate the electrochemical performance of supercapacitors. And the thickness of electrodes should be provided in the Methods section.*
6. *In Figure 2b, the green line cannot be found, the authors should clearly check the figures.*
7. *In Figure 3h, the scale range of x-axis and y-axis is inconsistent. In addition, have the EIS data been normalized by mass or surface are of electrode?*
8. *In page 13 of main text and Figure 4d, the author mentioned the energy density and power densities. The reviewer wonders that are these values calculated based on total mass of cells or only of electrode.*

We thankfully acknowledge all the reviewers for their time and valuable comments on our manuscript. The revisions have been highlighted in yellow in the revised manuscript. Please find below our point-by-point responses.

RESPONSE TO REVIEWERS' COMMENTS

Reviewer #1:

The authors report a 4D printing approach for manufacturing MXene hydrogels. This strategy shows remarkable universality, which suits not only the most commonly studied $Ti_3C_2T_x$ MXene but also Nb_2CT_x and $Mo_2Ti_2C_3T_x$. These three MXenes with different atomical structures and properties can represent a family of MXenes. It has never been achieved in previous reports. Besides, the printed MXene hydrogel architectures are precise and customizable, which is unlikely to realize in traditional model-dependent methods. The formation mechanism of MXene hydrogel is also clearly investigated. In capacitive application, the thickness/mass loading-independent rate capability of MXene hydrogel is very impressive, which surpasses the most advanced electrodes ever reported. The areal capacitance of MXene hydrogel also meets the commercial requirements even at a high scan rate of 1 V s^{-1} . The further printed micro-supercapacitor delivers outstanding capacitive performance and prominent low-temperature tolerance. This work is very interesting, which will advance the development of both MXenes and hydrogels and attract a wide range of attention in different fields, including additive manufacturing, material science, bioengineering, electrochemistry, etc. It is recommended to be published on Nature Communications after minor revision. Followings are some comments:

Response: We sincerely appreciate the reviewer for these positive comments and insightful suggestions, which have led to a further improvement of our manuscript.

1. The electrical conductivity of $Ti_3C_2T_x$ hydrogel is much higher than the other two MXene hydrogels, the authors should discuss the possible reasons, and present the electrical conductivity values of three MXene films for comparison. In addition, electrical conductivity is crucial in various applications, such as biomedical, sensing, and electrochemistry. The comparison of the

electrical conductivity of the MXene hydrogels with other reports should be presented, as well as their application fields. This will reveal the other potential applications of MXene hydrogels.

Response: We have measured the electrical conductivity of three MXene films in the revised version (Figure R1 (Supplementary Figure 13)). The electrical conductivities of Nb_2CT_x film, $\text{Ti}_3\text{C}_2\text{T}_x$ film, and $\text{Mo}_2\text{Ti}_2\text{C}_3\text{T}_x$ film are 382 S m^{-1} , $58,149 \text{ S m}^{-1}$, and $3,018 \text{ S m}^{-1}$, respectively. These values are an order of magnitude higher than their corresponding hydrogels, suggesting that the electrical conductivity of MXene hydrogels is highly dependent on the electrical conductivity of MXenes. The $\text{Ti}_3\text{C}_2\text{T}_x$ film has the highest conductivity and endows the $\text{Ti}_3\text{C}_2\text{T}_x$ hydrogel with the highest conductivity. Conversely, Nb_2CT_x hydrogel shows the lowest conductivity. In the revised manuscript, we have added a statement to explain the electrical conductivity difference between the three MXene hydrogels: “The conductivity difference between these hydrogels should be ascribed to the intrinsic conductivity difference of the three MXenes (Supplementary Fig. 13).” (Lines 173-174, Page 6)

Figure R1 (Supplementary Figure 13). (a) Photograph of electrical conductivity test for MXene films. Two silver wires were attached to MXene film using silver paste. (b) I-V curves of MXene films. The electrical conductivities of Nb_2CT_x film, $\text{Ti}_3\text{C}_2\text{T}_x$ film, and $\text{Mo}_2\text{Ti}_2\text{C}_3\text{T}_x$ film are 382 S m^{-1} , $58,149 \text{ S m}^{-1}$, and $3,018 \text{ S m}^{-1}$, respectively. These values are about an order of magnitude higher than their corresponding hydrogels, suggesting that the electrical conductivity of MXenes determines the electrical conductivity of MXene hydrogels.

Following the reviewer's suggestion, we have created a new table (Supplementary Table 2) to list the electrical conductivity and application fields of various hydrogel materials. The electrical conductivity values of our 4D-printed MXene hydrogels are higher than most hydrogels, revealing the potential of our MXene hydrogels for wide applications, such as sensors, bioelectronics, electromagnetic interference shielding, and electrochemical energy storage. (Lines 171-173, Page 6)

Supplementary Table 2. Comparison of the electrical conductivity of various hydrogels and their application fields

Hydrogels	Electrical conductivity (S m ⁻¹)	Applications	Reference
4D printed Nb ₂ CT _x hydrogel	37	NA	This work
4D printed Ti ₃ C ₂ T _x hydrogel	1548	Supercapacitors	
4D printed Mo ₂ Ti ₂ C ₃ T _x hydrogel	231	NA	
Ti ₃ C ₂ T _x hydrogel	220	Solar steam generation	22
Ti ₃ C ₂ T _x /PVA hydrogel	~0.1	Capacitive deionization	23
Ti ₃ C ₂ T _x /PVA hydrogel	0.056	Sensors	24
Ti ₃ C ₂ T _x /poly(acrylamide-acrylic acid)/chitosan hydrogel	1.34	Sensors	25
Ti ₃ C ₂ T _x /cellulose nanofibrils/PAM hydrogel	0.19	Sensors	26
Ti ₃ C ₂ T _x /PAM/poly(N-isopropyl acrylamide) hydrogel	1.092	Sensors	27
Ti ₃ C ₂ T _x /PAM/PVA hydrogel	~0.06	Sensors	28
Ti ₃ C ₂ T _x /PAA hydrogel	3.8	Sensors	29
Ti ₃ C ₂ T _x /PAA hydrogel	0.8	Electromagnetic interference shielding	30
Ti ₃ C ₂ T _x /bacterial cellulose hydrogel	0.07	Wound healing	31
Ti ₃ C ₂ T _x /graphene/CNT hydrogel	175	Supercapacitors	32
PEDOT:PSS hydrogel	880	Supercapacitors	33
PEDOT:PSS hydrogel	10	Bioelectronics	34
PEDOT:PSS/PAA hydrogel	23	Bioelectronics	35
PEDOT:PSS/polypyrrole hydrogel	867	Biosensors	36
PEDOT:PSS/γ-polyglutamic acid hydrogel	12.5	Sensors	37
PEDOT:PSS/PAM/laponite hydrogel	26	Bioengineering	38
PEDOT:PSS/graphene oxide hydrogel	829.6	Bioelectronics	39
PEDOT:PSS/gold nanoparticle hydrogel	67000	Bioelectronics	40
PAA/polyaniline hydrogel	0.141	Bioengineering	41
P(urea-ionic liquid-3-sulfopropyl methacrylate potassium salt) hydrogel	3	Conductors	42
PVA/Aramid nanofibers/silver nanowire hydrogel	16600	Electromagnetic interference shielding	43
PAM/alginate/silver flake hydrogel	37400	Soft electronics	44

Graphene hydrogel	58	NA	45
Graphene hydrogel	0.5	Supercapacitors	46

Note: all the data are extracted from the reported values or figures in references.

PAM: polyacrylamide; PAA: Polyacrylic acid.

2. How about the capacitive performance of pure PEDOT:PSS hydrogel and $Ti_3C_2T_x$ MXene film?

Response: In the revised version, we have added the CV results of pure PEDOT:PSS hydrogel and $Ti_3C_2T_x$ MXene film. As shown in Figure R2 (Supplementary Figure 16), PEDOT:PSS hydrogel is an electrical double-layer capacitor electrode with a specific capacitance of 43.2 F g^{-1} , whilst $Ti_3C_2T_x$ film shows strong pseudocapacitive peaks and delivers much higher specific capacitance of 281.8 F g^{-1} , suggesting that the capacitance of 4D-printed hydrogel electrode is mainly from $Ti_3C_2T_x$. A statement is added in the revised manuscript: “PEDOT:PSS hydrogel and $Ti_3C_2T_x$ film were also tested for comparison, Supplementary Fig. 16.” (Lines 239-240, Page 9)

Figure R2 (Supplementary Figure 16). CV curves of PEDOT:PSS hydrogel and $Ti_3C_2T_x$ film at a scan rate of 10 mV s^{-1} . Their mass loadings are $\sim 1.5 \text{ mg cm}^{-2}$. The specific capacitances of $Ti_3C_2T_x$ film and PEDOT:PSS hydrogel are 281.8 F g^{-1} and 43.2 F g^{-1} , respectively.

3. In the Introduction section (Line 58), the authors claim that the 4D printing technology is scalable, but only several architectures are shown in this manuscript. The authors should manufacture more hydrogel units to demonstrate this property.

Response: In the revised version, we have added additional printed patterns. As shown in Figure R3 (Supplementary Figure 8), up to 16 $Mo_2Ti_2C_3T_x$ hydrogel MSC units were printed in 0.5 h, which further transformed from sols into hydrogels after a self-assembly process. This process is

consistently repeatable, more hydrogel patterns or architectures can be easily manufactured on a larger scale. (Line 149, Page 6)

Figure R3 (Supplementary Figure 8). Scale-up 4D-printing of $\text{Mo}_2\text{Ti}_2\text{C}_3\text{T}_x$ hydrogel MSC units (size $2.2 \text{ cm} \times 1.7 \text{ cm} \times 0.05 \text{ cm}$ (L \times W \times H)). 16 MSC units were 3D printed in 0.5 h, which further transformed from sols into hydrogels after a self-assembly process. This process is consistently repeatable, more hydrogel units can be easily manufactured.

4. Five kinds of architectures are shown in Figure 2e, but only 2 corresponding videos are presented in Supplementary Movies. It would be better to show all the relating videos.

Response: In the revised version, the videos that correspond to the photos shown in Figure 2e are provided as Supplementary Movies 1-5. (Line 144, Page 5)

5. The micro-supercapacitors exhibit excellent low-temperature tolerance and the PVE-EG- H_2SO_4 electrolyte is a key contributor. How about the ionic conductivity of this electrolyte in comparison with other similar electrolytes? This will advise the future development of low-temperature devices.

Response: In the revised version, we have created a new table (Supplementary Table 8) to compare the ionic conductivity of various gel electrolytes at different temperatures. The ionic conductivity of PVE-EG- H_2SO_4 is higher than most gel electrolytes, and this value could be further enhanced if some optimization is employed, such as introducing glycerol. (Line 350, Page 14)

Supplementary Table 8. Comparison of ionic conductivity of various low-temperature tolerant gel electrolytes

Electrolyte	Temperature (°C)	Ionic conductivity (mS cm ⁻¹)	Reference
PVA-EG-H ₂ SO ₄	25	67	This work
	0	58.5	
	-20	27.5	
PAM-EG-LiCl	20	~13.6	10
	0	~11.6	
	-20	~6.2	
	-40	2.38	
PVA-EG-glycerol-H ₂ SO ₄	20	~127	9
	0	~81	
	-20	~53	
	-30	16	
PVA-EG-Zn(Tf) ₂	RT	15.03	77
	-20	9.05	
	-40	3.53	
PVA-H ₃ PO ₄	0	1.25	14
PVA-KOH	25	97	78
	0	89	
	-20	60	
PAM-EG-H ₂ SO ₄	-30	13	13
PAM-PVP-H ₃ PO ₄	30	97	79
	0	~70	
	-40	49	
P(AM-co-DMAEMA)-AMP-gelatin-LiCl	25	13.6	8
	0	~8.1	
	-20	~7.4	
	-40	4.3	
PAMAA-chitosan-NaCl-LiSO ₄	RT	48	12
	-20	36	
PVA-PAMAA-glycerol-NaCl	-20	13.14	80
PAMPS-PAM-DMSO-LiCl	-20	8.2	81
PIP13FSI- PYR14FSI-SiO ₂	20	5.5	7
	0	~2	
	-20	~0.8	
Poly(vinylidene fluoride-co-hexafluoropropylene)-EMITf-Al(Tf) ₃	RT	~1.6	82
	-20	~0.8	

Note: all the data are extracted from the reported values or figures in references.

PAM: polyacrylamide; PAA: Polyacrylic acid; PVP: Polyvinylpyrrolidone; P(AM-co-DMAEMA): poly(acrylamide-co-2-(dimethylamino)ethylmethacrylate); AMP: adenosine monophosphate; PAMAA: poly(acrylic amide-acrylic acid); PAMPS: poly(2-acrylamido-2-methylpropane sulfonic acid)

6. Some minor errors are found, such as “10 mins” (Line 399) and “20 min” (Line 405). Please carefully check the manuscript.

Response: We have carefully checked the whole manuscript and corrected typos and errors, including the mentioned one. (Line 418, Page 17)

Reviewer #2:

In this manuscript, the author shows out a universal 4D printing technology for manufacturing MXene HGs with customizable geometries and obtain an excellent electrochemical performance in supercapacitors. Besides, the low-temperature performance of micro-supercapacitors is also tested. But in my opinion, this manuscript has the defects: (1) the novelty is not enough for publication in Nature Commination; (2) compared to 3D printing technology or other methods, the advantages of 4D printing technology are not highlighted. Therefore, I think that this manuscript should be major revised or published others. Following are some comments that are concerned:

Response: The authors would like to thank the reviewer for these valuable and insightful comments. Following the comments, we have added more discussion and data to address all the raised suggestions.

(1) For the novelty concerns, our work shows great advances over previous reports and realizes some achievements that have never or hardly been achieved before:

Firstly, the previous reports all focused on $Ti_3C_2T_x$; no other MXene hydrogels (beyond $Ti_3C_2T_x$) have been reported. In this work, three kinds of MXene hydrogels (Nb_2CT_x , $Ti_3C_2T_x$, and $Mo_2Ti_2C_3T_x$) are manufactured. These three kinds of MXenes possess different numbers of atomic layers ($n=1, 2, \text{ and } 3$) and different transition metals (Nb, Ti, Mo) on their surface, representing a large family of MXenes with diverse structures and properties. The successful manufacturing of these three MXene hydrogels demonstrates the excellent versatility of our 4D printing technology and suggests its potential for producing more kinds of MXene hydrogels.

Secondly, the polymer crosslinkers employed in other reports were insulating, which lowered the electrical conductivity of MXene hydrogels and weakened their electrical/electrochemical performance. In our work, the conducting polymer PEDOT:PSS is employed and it shows strong

interactions with MXenes. As a result, our MXene hydrogels show higher electrical conductivity than most hydrogels, revealing the potential of our MXene hydrogels for wide applications, such as sensors, bioelectronics, electromagnetic interference shielding, and electrochemical energy storage (Supplementary Table 2). (Lines 171-173, Page 6)

Supplementary Table 2. Comparison of the electrical conductivity of various hydrogels and their application fields

Hydrogels	Electrical conductivity (S m ⁻¹)	Applications	Reference
4D printed Nb ₂ CT _x hydrogel	37	NA	This work
4D printed Ti ₃ C ₂ T _x hydrogel	1548	Supercapacitors	
4D printed Mo ₂ Ti ₂ C ₃ T _x hydrogel	231	NA	
Ti ₃ C ₂ T _x hydrogel	220	Solar steam generation	22
Ti ₃ C ₂ T _x /PVA hydrogel	~0.1	Capacitive deionization	23
Ti ₃ C ₂ T _x /PVA hydrogel	0.056	Sensors	24
Ti ₃ C ₂ T _x /poly(acrylamide-acrylic acid)/chitosan hydrogel	1.34	Sensors	25
Ti ₃ C ₂ T _x /cellulose nanofibrils/PAM hydrogel	0.19	Sensors	26
Ti ₃ C ₂ T _x /PAM/poly(N-isopropyl acrylamide) hydrogel	1.092	Sensors	27
Ti ₃ C ₂ T _x /PAM/PVA hydrogel	~0.06	Sensors	28
Ti ₃ C ₂ T _x /PAA hydrogel	3.8	Sensors	29
Ti ₃ C ₂ T _x /PAA hydrogel	0.8	Electromagnetic interference shielding	30
Ti ₃ C ₂ T _x /bacterial cellulose hydrogel	0.07	Wound healing	31
Ti ₃ C ₂ T _x /graphene/CNT hydrogel	175	Supercapacitors	32
PEDOT:PSS hydrogel	880	Supercapacitors	33
PEDOT:PSS hydrogel	10	Bioelectronics	34
PEDOT:PSS/PAA hydrogel	23	Bioelectronics	35
PEDOT:PSS/polypyrrole hydrogel	867	Biosensors	36
PEDOT:PSS/γ-polyglutamic acid hydrogel	12.5	Sensors	37
PEDOT:PSS/PAM/laponite hydrogel	26	Bioengineering	38
PEDOT:PSS/graphene oxide hydrogel	829.6	Bioelectronics	39
PEDOT:PSS/gold nanoparticle hydrogel	67000	Bioelectronics	40
PAA/polyaniline hydrogel	0.141	Bioengineering	41
P(urea-ionic liquid-3-sulfopropyl methacrylate potassium salt) hydrogel	3	Conductors	42
PVA/Aramid nanofibers/silver nanowire hydrogel	16600	Electromagnetic interference shielding	43
PAM/alginate/silver flake hydrogel	37400	Soft electronics	44
Graphene hydrogel	58	NA	45
Graphene hydrogel	0.5	Supercapacitors	46

Note: all the data are extracted from the reported values or figures in references.

PAM: polyacrylamide; PAA: Polyacrylic acid.

Thirdly, the geometry of the previously reported MXene hydrogels strongly depends on the shape and size of molds, which is unlikely to meet the requirements of complexity and precision in many scenarios, especially in the context of the rapid development of portable electronics. In this work, our 4D printing technology allows the precise and mold-free fabrication of various complex MXene hydrogels, such as microlattice, rectangular hollow prism, Chinese knot, “CRANN” logo, and micro-supercapacitor (MSC) units. And series of substrates suit, e.g., glass slide, cloth, and PET film (Figure R4 (Figure 2e)). This 4D printing technology significantly surpasses the mold-dependent methods and shows great potential for modern electronics.

Figure R4 (Figure 2e). Photographs of 4D-printed MXene hydrogel architectures (from left to right): $\text{Ti}_3\text{C}_2\text{T}_x$ hydrogel microlattice on glass slide, $\text{Ti}_3\text{C}_2\text{T}_x$ hydrogel rectangular hollow prism on glass slide, Nb_2CT_x hydrogel Chinese knot on cloth, Nb_2CT_x hydrogel “CRANN” logo on PET film, flexible $\text{Mo}_2\text{Ti}_2\text{C}_3\text{T}_x$ hydrogel MSC units on PET film. All scale bars correspond to 1 cm.

Fourthly, benefiting from the large specific surface area, high electrical conductivity, and hydrophilic property, ultrahigh capacitance (3.32 F cm^{-2} at 10 mV s^{-1} and 233 F g^{-1} at 10 V s^{-1}) and unprecedented mass loading/thickness-independent rate capabilities are achieved for our $\text{Ti}_3\text{C}_2\text{T}_x$ hydrogel electrode, which surpasses most state-of-the-art electrode materials (Supplementary Table 5). Particularly, in commercial applications, the typical electrode areal capacitance is 0.6 F cm^{-2} (horizontal dash line in Figure R5 (Figure 3g)), and to date, only the liquid-crystalline $\text{Ti}_3\text{C}_2\text{T}_x$ (*Nature* **557**, 409-412 (2018)) and 1T-MoS_2 (*Nat. Nanotech.* **17**, 153-158 (2022)) electrodes have achieved this value at 1 V s^{-1} . Our MXene hydrogel electrodes not only meet the commercial requirements, but also possess X2 higher areal capacitance than liquid-crystalline $\text{Ti}_3\text{C}_2\text{T}_x$ and 1T-MoS_2 electrodes at both 1 and 2 V s^{-1} under similar mass loadings ($6.16\text{--}6.6 \text{ mg cm}^{-2}$) (Figure R5 (Figure 3g)). This is a very large progress and demonstrates the potential of our MXene hydrogels for practical applications. The further fabricated MSCs also

deliver larger areal capacitance and higher energy/power densities than most printed MSCs, including the 3D-printed MXene MSCs. The low-temperature performance of our MSC is excellent as well (Figure R6 (Figure 4c,d,f)).

Supplementary Table 5. Comparison of electrochemical performance of various high-rate supercapacitor electrodes

Electrode	Mass loading (mg cm ⁻²)	Areal capacitance at 10 mV s ⁻¹	Capacitance retention from 10 to 1,000 mV s ⁻¹	Specific capacitance at 10 V s ⁻¹	Reference
4D printed Ti ₃ C ₂ T _x hydrogel	0.5	0.14 F cm ⁻²	96.3%	232.9 F g ⁻¹	This work
	3.1	0.86 F cm ⁻²	92.4%	104.4 F g ⁻¹	
	6.6	1.83 F cm ⁻²	90.2%	51.8 F g ⁻¹	
	11.8	3.32 F cm ⁻²	26.7%	NA	
Filtered porous Ti ₃ C ₂ T _x film	0.53	0.1378 F cm ⁻²	87.3%	207.9 F g ⁻¹	49
Ti ₃ C ₂ T _x /NbN film	1.05	0.29 F cm ⁻²	71%	78.8 F g ⁻¹	50
Ti ₃ C ₂ T _x /rGO hydrogel	1.16	0.3423 F cm ⁻²	81%	NA	51
Filtered Ti ₃ C ₂ T _x film	2.1	NA	43.6%	NA	52
Wavy Ti ₃ C ₂ T _x film	2.28	0.7662 F cm ⁻²	88.9%	NA	
Oxidized Ti ₃ C ₂ T _x film	1	0.34 F cm ⁻²	68.2%	71 F g ⁻¹	53
	2.3	0.73 F cm ⁻²	41.5%	27 F g ⁻¹	
	4.4	1.35 F cm ⁻²	28%	20 F g ⁻¹	
	9.1	2.52 F cm ⁻²	22.6%	13 F g ⁻¹	
	12	3.0 F cm ⁻²	22%	9.5 F g ⁻¹	
Filtered Ti ₃ C ₂ T _x hydrogel	1.2	0.42 F cm ⁻²	59%	37.8 F g ⁻¹	54
	5.3	1.84 F cm ⁻²	13.5%	NA	
	11.3	3.7 F cm ⁻²	12.5%	NA	
Porous Ti ₃ C ₂ T _x	0.43	0.14 F cm ⁻²	92.8%	210 F g ⁻¹	
	0.9	0.27 F cm ⁻²	87%	121.6 F g ⁻¹	
	4.3	1.33 F cm ⁻²	41%	31 F g ⁻¹	
Liquid-crystal Ti ₃ C ₂ T _x	2.8	0.76 F cm ⁻²	80.3%	126.2 F g ⁻¹	55
	3.6	0.93 F cm ⁻²	65.4%	98.4 F g ⁻¹	
	6.16	1.53 F cm ⁻²	42.8%	21.4 F g ⁻¹	
1T-MoS ₂ film	6.64	1.38 F cm ⁻²	53.6%	31 F g ⁻¹	56
	10.87	2.01 F cm ⁻²	37.3%	12 F g ⁻¹	
Graphene ribbon film	1	NA	53.2%	NA	57
Ti ₃ C ₂ T _x hydrogel	NA	NA	87%	NA	58
Ti ₃ C ₂ T _x hydrogel	NA	NA	50.3%	NA	59

Note: all the data are extracted from the reported values or figures in references.

Figure R5 (Figure 3g). Comparison of the areal capacitance of $\text{Ti}_3\text{C}_2\text{T}_x$ hydrogels with benchmark electrodes at scan rates of 1,000 and 2,000 mV s^{-1} . These electrodes are porous $\text{Ti}_3\text{C}_2\text{T}_x$ ¹², liquid-crystalline $\text{Ti}_3\text{C}_2\text{T}_x$ ⁴¹, 1T-MoS₂⁴², filtered $\text{Ti}_3\text{C}_2\text{T}_x$ hydrogel¹², and filtered porous $\text{Ti}_3\text{C}_2\text{T}_x$ ⁴³.

Figure R6 (Figure 4c,d,f). (Figure 4c) Comparison of the areal capacitance of 4D-printed $\text{Ti}_3\text{C}_2\text{T}_x$ hydrogel MSC with other printed MSCs. (Figure 4d) Ragone plots of 4D-printed $\text{Ti}_3\text{C}_2\text{T}_x$ hydrogel MSC and other high-performance MSCs. (Figure 4f) Capacitance retention of 4D-printed $\text{Ti}_3\text{C}_2\text{T}_x$ hydrogel MSC during cooling/heating cycles.

To conclude, our work has addressed many difficulties that have not been solved and made great achievements over previous reports in both MXene hydrogels manufacturing and electrochemical energy storage. We think our work is novel and has met the high level of Nature Communications.

(2) To compare the 4D printing with 3D printing and highlight the advantages of 4D printing, we have added a new paragraph in the Introduction section: “Additive manufacturing, or 3D printing, offers an efficient approach to realizing the precise, mold-free, and low-cost fabrication

of complex objects by layer-by-layer deposition of material²¹. With the introduction of the fourth dimension of time, 4D printing (3D printing + time) emerged²². It not only inherits all merits of 3D printing but also allows the static objects created by 3D printing to change their shape, property, or functionality over time when exposed to specific external stimuli (*e.g.*, heat, light, water, pH)²³, endowing the printed objects with new features. However, no related works on MXene hydrogels were ever reported.” (Lines 57-63, Pages 2-3)

1. Compared to 3D printing technology, what are the advantages of MXene HGs by 4D printing? The authors should give more discussion in the introduction.

Response: The 3D-printed MXene patterns or architectures are usually in sol states, which are essentially inks or condensed dispersions but with customizable geometries (*e.g.*, *Adv. Mater.* **31**, 1902725 (2019); *ACS Nano* **14**, 640-650 (2019)). Once immersed in water, they will be easily re-dispersed. Therefore, freeze-drying is indispensable to maintain the shape of 3D-printed objects before application. Besides, the final products are aerogels, not hydrogels. In contrast, our 4D-printed MXene hydrogels are real hydrogels with ~95.8 wt.% water inside (the solid contents of MXene hydrogels are ~4.2 wt.%, Line 478, Page 19), which will facilitate the electrolyte ion transport and maximize the electrochemical performance of MXene electrodes. Moreover, our MXene hydrogels show good mechanical strength and can stay stable even after shaking, which can act as electrodes directly. This property will also benefit our MXene hydrogels in other application fields, such as sensors and bioelectronics.

To highlight the advantages of our work, we have revised a sentence in Introduction and created a new table (Supplementary Table 1) in Supplementary Information. Now the difference and advantages of 4D-printed MXene hydrogels are more clearly presented: “Differing from the dissolvable MXene sol patterns produced by traditional 3D printing (Supplementary Table 1), in our 4D printing technology, crosslinked MXene hydrogels with enhanced mechanical strengths are obtained by employing a simple heat-stimulated self-assembly process.” (Lines 66-69, Page 3)

Supplementary Table 1. Comparison of 4D printing technology with traditional 3D printing on producing 3D MXene architectures

Printing technology	Ink formulation	Products	Properties	Reference
4D printing	MXenes (Nb_2CT_x , $\text{Ti}_3\text{C}_2\text{T}_x$, or $\text{Mo}_2\text{Ti}_2\text{C}_3\text{T}_x$) + PEDOT:PSS + DMSO + H_2SO_4 + Sodium L-ascorbate	Nb_2CT_x hydrogel	 1. Hydrogel state. 2. Customizable geometries. 3. Strong interactions between MXenes and polymer chains. 4. Satisfying mechanical strength and operability. 5. Stable in water. 6. hydrogels can be used as electrodes directly. 	This work
		$\text{Ti}_3\text{C}_2\text{T}_x$ hydrogel		
		$\text{Mo}_2\text{Ti}_2\text{C}_3\text{T}_x$ hydrogel		
3D printing	$\text{Ti}_3\text{C}_2\text{T}_x$	$\text{Ti}_3\text{C}_2\text{T}_x$ sol	 1. Sol/ink state. 2. Customizable geometries. 3. Weak interactions between MXene flakes, or MXenes and other components. 4. Unstable in water (re-dispersible). 5. Freeze-drying is essential to maintain the shape of 3D printed patterns before conducting electrochemical tests. 	4, 5, 16, 17
3D printing	$\text{Ti}_3\text{C}_2\text{T}_x$ + $\text{ZnSO}_4 \cdot 7\text{H}_2\text{O}$	$\text{Ti}_3\text{C}_2\text{T}_x$ sol		18
3D printing	$\text{Ti}_3\text{C}_2\text{T}_x$ + cellulose nanofiber	$\text{Ti}_3\text{C}_2\text{T}_x$ -cellulose nanofiber sol		19
3D printing	V_2CT_x + CNT + GO	V_2CT_x -CNT-GO sol		20
3D printing	$\text{NiCoP}/\text{Ti}_3\text{C}_2\text{T}_x$ + CNT	$\text{NiCoP}/\text{Ti}_3\text{C}_2\text{T}_x$ -CNT sol		21

Additionally, we also provide videos and photographs of 4D-printed $\text{Ti}_3\text{C}_2\text{T}_x$ hydrogels and 3D-printed $\text{Ti}_3\text{C}_2\text{T}_x$ sol before and after shaking (Supplementary Movies 6-7, Figures R7-R8 (Supplementary Figures 9-10)) in the revised manuscript. The 3D-printed $\text{Ti}_3\text{C}_2\text{T}_x$ sol microlattice broke into fragments despite the existence of some electrostatic attractions between the negatively charged MXenes and positively charged PEDOT⁺ chains and protons. Pure $\text{Ti}_3\text{C}_2\text{T}_x$ sol architectures (e.g., *Adv. Mater.* **31**, 1902725 (2019); *ACS Nano* **14**, 640-650 (2019)) that only possess weak van der Waals interactions will be completely redispersed in water after shaking. In contrast, our 4D-printed $\text{Ti}_3\text{C}_2\text{T}_x$ hydrogels show great integrity and stay stable after shaking, exhibiting much better mechanical strength and operability than 3D-printed objects. (Lines 150-155, Page 6)

Figure R7 (Supplementary Figure 9). Photographs of 4D-printed $\text{Ti}_3\text{C}_2\text{T}_x$ hydrogel microlattice and rectangular hollow prism before and after shaking for ~ 14 s. After shaking, the two hydrogels retained their integrity.

Figure R8 (Supplementary Figure 10). Photographs of 3D-printed $\text{Ti}_3\text{C}_2\text{T}_x$ sol microlattice before and after shaking for ~ 8 s. After shaking, the sol microlattice broke into fragments. It is worth noting that, there are already some electrostatic attractions between the negatively charged MXenes and positively charged PEDOT^+ chains and protons, which protect this sol microlattice from complete redispersion in water. The pure $\text{Ti}_3\text{C}_2\text{T}_x$ sol architectures that only possess weak van der Waals interactions^{4,5} will be completely redispersed after shaking.

2. The micro-supercapacitor delivers ultrahigh energy and power densities up to $93 \mu\text{Wh cm}^{-2}$ and 7 mW cm^{-2} , but compared to the previous literature, the power density value of this work cannot reach an ultrahigh value (e.g. *Adv. Energy Mater.* 2020, 10, 1903794; *Energy Environ. Sci.*, 2019, 12, 96-115).

Response: We agree with the reviewer's point. To make this expression more precise, we have replaced the word "ultrahigh" with "high". (Line 25, Page 1)

We read these two papers carefully and there are two main reasons that lead to higher power density than ours. Firstly, our device has a thickness of 4 mm; the devices in the mentioned papers are, in contrary, very thin (almost all of them are less than 0.5 mm). Secondly, our device is current-collector free; metal current collectors or high-conductive metal fillers (*e.g.*, Ag nanowire and Au) were employed in the cited papers.

3. In the “probing self-assembly mechanism” section, the authors mentioned that “suggesting the conformation change from benzene structure to quinoid structure and thus elongated conjugation lengths of PEDOT⁺ chains.” The authors are suggested to add necessary citation to rationalize the discussion.

Response: A reference is now cited after this sentence. (Line 182, Page 7)

4. The electrical conductivity of Ti₃C₂T_x HGs can reach 1548 S m⁻¹, which is far more than the cases of Nb₂C_{Tx} and Mo₂Ti₂C₃T_x HGs. The authors should give detailed reasons in the manuscript.

Response: The observation is highly appreciated and to amend this issue, we measured the electrical conductivity of three MXene films in the revised version and found the electrical conductivity of MXene hydrogels is highly dependent on the electrical conductivity of MXenes. To avoid duplication, please see our response to Reviewer #1’s 1st comment and Figure R1 (Supplementary Figure 13) on page 4.

5. More experimental details could be included in the manuscript. For instance, the electrolyte amount used in the devices should be provided, which is an important parameter to evaluate the electrochemical performance of supercapacitors. And the thickness of electrodes should be provided in the Methods section.

Response: In the revised version, we have added more experimental details:

(1) In the Methods section, all the known purities and manufacturers of chemicals are included: Nb₂AlC MAX (400 mesh, 98%, Jilin 11 Technology, Co. Ltd, China); LiF (Alfa Aesar, 97%); HCl (Sigma Aldrich, 37%); tetramethylammonium hydroxide (TMAOH) solution (Sigma Aldrich, 25 wt.%); Ti₃AlC₂ MAX (400 mesh, Carbon, Ukraine); HF (Acros Organics, 48–50

wt.%); PEDOT:PSS suspension (PH 1000, Clevios™); DMSO (Sigma Aldrich, 99.9%, anhydrous), H₂SO₄ (Honeywell, 95-97%); sodium L-ascorbate (Acros Organics, 99%); EG (Hach); PVA (Aldrich, Mw 89,000-98,000, 99%, hydrolyzed); activated carbon (YP-50, Kuraray, Japan) (Pages 17-19)

(2) The photograph, schematic, and model number (Line 500, Page 19) of the Swagelok cell that used for three-electrode tests are provided (Figure R9 (Supplementary Figure 15)).

Figure R9 (Supplementary Figure 15). (a) Photograph and (b) schematic of Swagelok cell for three-electrode test.

(3) The schematic of designing 4D-printed Ti₃C₂T_x hydrogel MSC is provided (Figure R10 (Supplementary Figure 21)).

Figure R10 (Supplementary Figure 21). Schematic of designing 4D-printed Ti₃C₂T_x hydrogel MSC. The volume percentage of electrodes to the whole MSC is ~70 vol.%, and the gap between electrodes is ~30 vol.%. Because the electrodes are highly porous, they can absorb almost as much electrolyte as their volume. Thus, the volume of the gel electrolyte added should be at least the same as the volume of MSC (including both the electrodes and the gap). To ensure the complete infiltration of electrodes and maximize the electrochemical performance of MSC, the total volume of the cast PVA-EG-H₂SO₄ gel electrolyte was set to 120 vol.% of the MSC.

(4) The size of MSC (including the electrode/MSM thickness) and the electrolyte volume are disclosed in Methods: 1.8 mL PVA-EG-H₂SO₄ gel electrolyte was added for a 2.2 cm × 1.7 cm × 0.4 cm (L × W × H) MSC, and 0.22 mL gel electrolyte was added for a 2.2 cm × 1.7 cm × 0.05 cm MSC. (Lines 488-496, Page 19); 0.5 mL 3 M H₂SO₄ was used in the Swagelok cell for the three-electrode test. (Line 502, Page 19)

6. In Figure 2b, the green line cannot be found, the authors should clearly check the figures.

Response: In the revised version, we have changed the colors and symbols of the three lines. Although the lines of Nb₂CT_x (red) and Ti₃C₂T_x (blue) inks are almost overlapped, the three samples are readable now (Figure R11 (Figure 2b)).

Figure R11 (Figure 2b). Viscosity as a function of shear rate for Nb₂CT_x, Ti₃C₂T_x and Mo₂Ti₂C₃T_x inks.

7. In Figure 3h, the scale range of x-axis and y-axis is inconsistent. In addition, have the EIS data been normalized by mass or surface area of electrode?

Response: (1) Usually, Nyquist plots should be represented in a one-to-one ratio, where the X-axis and Y-axis form a square plot (*Adv. Energy Mater.* **9**, 1902007 (2019)). In the grey square area of Figure 3h, the x-axis and y-axis have the same scale range of 0 to 0.65 Ω cm², which meets the law for EIS data presentation. The proportional enlargement of the ordinate of Figure 3h does not change this but allows the alignment of Figure 3h with other figures in Figure 3. This data presentation method has also been widely used in many other papers, such as *Science* **341**, 1502-1505 (2013); *Nat. Energy* **2**, 17105 (2017); *Nat. Commun.* **10**, 1795 (2019).

Figure R12 (Figure 3h). EIS plots of $\text{Ti}_3\text{C}_2\text{T}_x$ hydrogels with different mass loadings taken at 0.2 V.

(2) EIS data have been normalized by surface area of electrode, as shown by the unit “ $\Omega \text{ cm}^2$ ”.

8. In page 13 of main text and Figure 4d, the author mentioned the energy density and power densities. The reviewer wonders that are these values calculated based on total mass of cells or only of electrode.

Response: The energy and power densities shown in Figure 4d are calculated based on the total area of the MSC (including both the electrodes and gap). This has been clarified in Supplementary Information as follows:

The areal energy density (mWh cm^{-2}) and average power density (mW cm^{-2}) of MSC were calculated based on the discharge scan of the GCD curves:

$$E_{MSC} = \frac{i}{S_{MSC}} \times \left(\int_0^{t_{MAX}} V_{MSC} \cdot dt \right)$$

and

$$P_{MSC} = \frac{E_{MSC}}{\Delta t}$$

Here, i is the current, S_{MSC} is the total area of MSC (including both the electrodes and gap), t_{MAX} is the maximum discharge time, V_{MSC} is the voltage window of MSC, Δt is the discharging time.

Reviewer comments, further round review

REVIEWERS' COMMENTS

Reviewer #1 (Remarks to the Author):

I think this manuscript could be accepted in this revised form. All my comments have been well responded.

Reviewer #2 (Remarks to the Author):

No questions.